# Disentangling Linguistic Features with Dimension-Wise Analysis of Vector Embeddings

## Abstract

Understanding the inner workings of neural embeddings, particularly in models such as BERT, remains a challenge because of their high-dimensional and opaque nature. This paper proposes a framework for uncovering the specific dimensions of vector embeddings that encode distinct linguistic properties (LPs). We introduce the Linguistically Distinct Sentence Pairs (LDSP-10) dataset, which isolates ten key linguistic features such as synonymy, negation, tense, and quantity. Using this dataset, we analyze BERT embeddings with various statistical methods, including the Wilcoxon signed-rank test, mutual information, and recursive feature elimination, to identify the most influential dimensions for each LP. We introduce a new metric, the Embedding Dimension Importance (EDI) score, which quantifies the relevance of each embedding dimension to a LP. Our findings show that certain properties, such as negation and polarity, are robustly encoded in specific dimensions, while others, like synonymy, exhibit more complex patterns. This study provides insights into the interpretability of embeddings, which can guide the development of more transparent and optimized language models, with implications for model bias mitigation and the responsible deployment of AI systems. [1]

## 1 Introduction

Word embeddings are central to natural language processing (NLP), enabling machines to represent and interpret text in continuous vector spaces. From early models like Word2Vec Mikolov et al. (2013) and GloVe Pennington et al. (2014), to advanced models like GPT-2 Radford et al. (2019) and BERT Devlin et al. (2019), embeddings have evolved to capture complex linguistic nuances. BERT, in particular, leverages bidirectional transformers to generate contextualized word representations, enhancing syntactic and semantic understanding Rogers et al. (2020).

Despite these advancements, embeddings are often seen as "black boxes," where the high-dimensional nature of the spaces they occupy makes interpretation difficult Belinkov & Glass (2019). The field of interpretable embeddings seeks to address these challenges by making the dimensions of embeddings more transparent and meaningful Faruqui et al. (2015a); Incitti et al. (2023); Snidaro et al. (2019). However, most systems still rely on popular embedding models like GPT, BERT, Word2Vec, and GloVe, which prioritize performance over interpretability Cao (2024); Lipton (2017).

Our research introduces a generalizable framework for identifying specific embedding dimensions in models like BERT and GPT-2 that encode distinct LPs. This work responds to the growing need for interpretable models, especially for tasks like bias mitigation Bolukbasi et al. (2016); Mehrabi et al. (2021), task-specific optimization Guyon & Elisseeff (2003); Voita et al. (2019), and more system controllability Bau et al. (2019).

We present the LDSP-10 dataset, which consists of sentence pairs isolating nine LPs, designed to probe embedding spaces and identify the dimensions most influential for each property. We analyze these sentence pairs using statistical tests, mutual information, and feature selection methods. We propose the **Embedding Dimension Importance** (EDI) score, which aggregates these analyses to quantify the relevance of each dimension to specific LPs.

---

[1]Code will be released upon publication.

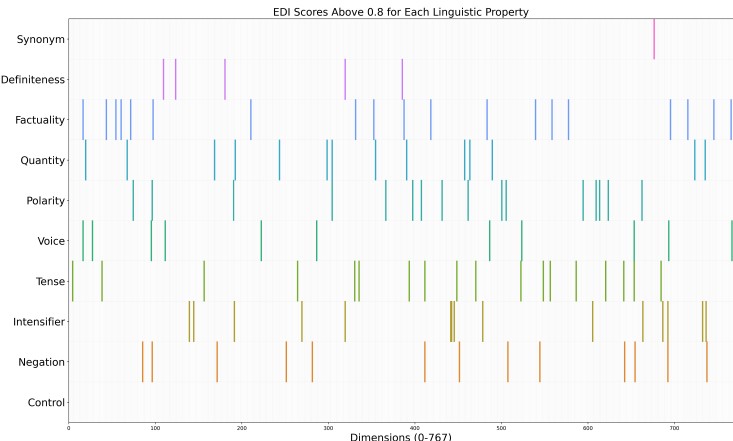

Figure 1: Dimensions of BERT embeddings that encode the most information about each LP. Relevance is determined by Embedding Dimension Importance (EDI) scores above 0.8, a threshold chosen in relation to the general EDI score distribution.

| | Control | Synonym | Quantity | Tense | Intensifier | Voice | Definiteness | Factuality | Polarity | Negation |
|---|---|---|---|---|---|---|---|---|---|---|
| BERT | 0.5033 | 0.7033 | 0.95 | 0.94 | 0.9867 | 0.9667 | 0.8967 | 0.9833 | 0.9700 | 0.9333 |
| GPT-2 | 0.57 | 0.6267 | 0.9733 | 0.9567 | 0.9367 | 0.9867 | 0.9433 | 0.9667 | 0.9533 | 0.93 |
| MP-Net | 0.54 | 0.5267 | 0.9533 | 0.93 | 0.8733 | 0.86 | 0.8567 | 0.9667 | 0.9533 | 0.9367 |

Table 1: Evaluation 1 (§ 5.2) accuracy for different LPs across BERT, GPT-2, and MP-Net. A simple logistic classifier is able to perform at these levels of accuracy on the highest EDI subset of dimensions of embeddings from each of these models.

This paper makes three contributions. First, is the introduction of the LDSP-10 dataset, consisting of sentence pairs that isolate nine LPs. Second is a generalizable framework and quantifiable metric (EDI score) for identifying influential embedding dimensions, applicable to different models and linguistic features. Third is a comprehensive analysis of BERT, GPT-2, and MPNet embeddings, revealing key dimensions related to each LP.

## 2 RELATED WORKS

Research on interpretable embeddings can be divided into two categories: interpretable embeddings and representation analysis. The former focuses on designing models that naturally produce interpretable representations, while the latter involves post-hoc analysis to uncover how existing embeddings encode human-interpretable features.

### 2.1 INTERPRETABLE EMBEDDINGS

Several approaches have been proposed to create interpretable word embeddings. Early efforts like Murphy et al. (2012) used matrix factorization techniques to generate sparse, interpretable embeddings. Faruqui et al. (2015b) introduced Sparse Overcomplete Word Vectors (SPOWV), which used a dictionary learning framework for more interpretable, sparse embeddings. Other methods, such as Guillot et al. (2023) and Subramanian et al. (2018), explored how sparsification techniques could disentangle properties within embeddings, making them more interpretable.

Approaches to embedding interpretability also involve aligning dimensions with human-understandable concepts. For instance, Panigrahi et al. (2019) used Latent Dirichlet Allocation (LDA) to produce embeddings where each dimension corresponds to a specific word sense, and Benara et al. (2024) employed LLM-powered yes/no question-answering techniques to generate interpretable embeddings. Despite these innovations, popular models like Word2Vec, GloVe, and BERT remain dominant in NLP but often lack inherent interpretability. As a result, methods for post-hoc analysis are needed to interpret these embeddings.

## 2.2 REPRESENTATION ANALYSIS

Representation analysis focuses on understanding how knowledge is structured within embeddings and how individual neurons contribute to encoding specific properties Sajjad et al. (2022). Senel et al. (2017) demonstrated how individual dimensions correspond to specific semantic properties, and Zhu et al. (2018) emphasized the value of sentence-level embeddings in capturing nuanced semantic properties. Research has also explored the linguistic features encoded within embeddings. Conneau et al. (2018) developed a set of ten probing tasks that evaluate how sentence embeddings capture various linguistic features, such as syntactic structures and semantic roles. Adi et al. (2017) complemented this work by proposing classification tasks that reveal the effectiveness of sentence embeddings in encoding attributes like sentence length and word order.

Recent research has analyzed individual neurons in embedding spaces, often using methods like neuron-ranking, where a probe is used to rank neurons based on their relevance to a specific linguistic feature Dalvi et al. (2019); Durrani et al. (2020); Torroba Hennigen et al. (2020). Antverg & Belinkov (2022) analyzed these methods, separating representational importance from functional utility and introducing interventions to evaluate whether encoded information is actively utilized.

Building on this foundation, Durrani et al. (2024) introduced Linguistic Correlation Analysis (LCA), which identifies salient neurons that encode specific linguistic features. Their findings indicated redundancy in information encoding across neurons, enhancing robustness in representation learning. Similarly, Gurnee et al. (2023) proposed sparse probing methods to address polysemanticity, illustrating how features are distributed across neurons in transformer models. Additionally, Torroba Hennigen et al. (2020) presented intrinsic probing, introducing a Gaussian framework to identify dimensions encoding LPs. We Together, these findings suggest that linguistic attributes are often encoded in focal dimensions, providing insights into how different models represent linguistic knowledge.

| Property | Sentence Pair |
|---|---|
| Control | They sound excited. The farmer has 20 sheep. |
| Synonym | The music was calming. The music was soothing. |
| Quantity | I ate two cookies. I ate several cookies. |
| Tense | The river flows swiftly. The river flowed swiftly. |
| Intensifier | The task is easy. The task is surprisingly easy. |
| Voice | The team won the game. The game was won by the team. |
| Definiteness | The bird flew away. A bird flew away. |
| Factuality | The car is red. The car could be red. |
| Polarity | She passed the exam. She failed the exam. |
| Negation | The project is successful. The project is not successful. |

Table 2: Sample linguistically distinct sentence pairs (LDSPs) from each of the LPs tested in this study. LDSP-10 dataset contains 1000 sentence pairs per LP. Control LDSPs are randomly chosen from the dataset, intended to be unrelated, as a baseline for our analysis.

Our work builds on these ideas by using the LDSP-10 dataset to isolate linguistic features, which provides a focused method for assessing how embedding dimensions capture these properties. We move beyond traditional probing and neuron-ranking techniques to offer a more targeted examination of embedding interpretability.

## 3 LINGUISTICALLY DISTINCT SENTENCE PAIRS (LDSP-10) DATASET

We curated a dataset of 1000 LDSPs for each of the 10 LPs we wanted to investigate. The dataset was generated using Google's `gemini-1.5-flash` model API. This model was selected due to its reliability and cost-efficiency while being able to produce consistent outputs across a variety of linguistic contexts. The model was prompted with a set of reference LDSPs as well as a description of the LP to ensure a high-quality outputs. These outputs were generated in batches of 100 LDSPs at a time. To ensure reproducibility and transparency, the detailed prompts used to generate the dataset are provided in Appendix A. These prompts included explicit examples of each LP, along with clear instructions tailored to the `gemini-1.5-flash` API to encourage outputs adhering to the desired properties.

During the dataset creation process, the order of the sentences in the LDSP was not always consistent with the intended property distinction. We made modifications to the prompt to explicitly enforce the correct ordering. This adjustment ensured that the generated outputs reliably aligned with our

expectations. Manual validation was conducted to assess the quality of the generated data. The evaluation revealed that more than 99% of the sampled sentence pairs adhered to the minimal distinctions expected for their LP. The system exhibited a low rate of syntactic or content biases, with errors occurring primarily in cases involving more complex distinctions, such as polarity and factuality.

The LPs tested were chosen to explore various semantic and syntactic relationships. We generated LDSPs for *definiteness*, *factuality*, *intensifier*, *negation*, *polarity*, *quantity*, *synonym*, and *tense*. In addition, we generated a *control* group, which contains sentence pairs of completely unrelated sentences. This is used to compare to the LDSPs and contextualize our observed results. Example LDSPs can be found in Table 2, with more detailed definitions found in Appendix B. For more information about the dataset generation pipeline, please refer to Appendix A.

# 4 DIMENSION-WISE EMBEDDING ANALYSIS

## 4.1 WILCOXON SIGNED-RANK TEST

The Wilcoxon signed-rank test is employed in our analysis to assess whether there exists a significant difference in embedding dimensions across paired sentence representations. This non-parametric test is particularly useful when the data does not conform to the normality assumptions required by parametric tests such as the paired t-test. Given that sentence embeddings often exhibit complex, non-Gaussian distributions, the Wilcoxon test provides a robust approach to evaluating the statistical significance of differences in embedding dimensions.

Formally, let $X_1, X_2 \in \mathbb{R}^d$ be the embedding representations of two paired sentences. We define the difference vector as:

$$D = X_1 - X_2,\tag{1}$$

where $D = \{d_1, d_2, ..., d_d\}$ contains the differences for each embedding dimension. The null hypothesis for the Wilcoxon test is given by:

$$H_0 : \text{median}(D) = 0,\tag{2}$$

which posits that there is no significant shift in the embedding dimensions between the two sentence representations.

The test proceeds by ranking the absolute values of the nonzero differences, assigning ranks $R_i$ to each $|d_i|$. The Wilcoxon test statistic $W$ is computed as the sum of ranks corresponding to positive differences:

$$W = \sum_{d_i > 0} R_i.\tag{3}$$

The significance of $W$ is then assessed using either critical values from the Wilcoxon distribution or by computing a $p$-value.

We employ the Wilcoxon test in our framework to analyze whether certain dimensions of the embeddings exhibit systematic shifts between sentence pairs. Overall, the Wilcoxon signed-rank test provides a rigorous statistical method for validating the role of embedding dimensions in differentiating sentence pairs, ensuring that our conclusions are drawn from statistically significant evidence rather than random variations.

## 4.2 MUTUAL INFORMATION (MI)

To further investigate the relationship between embedding dimensions and each LP and inspired by Pimentel et al. (2020), we employed mutual information (MI) analysis. Mutual information is a measure of the mutual dependence between two variables, quantifying the amount of information obtained about one variable by observing the other Zeng (2015).

For discrete random variables $X$ and $Y$, the mutual information $MI(X;Y)$ is defined as:

$$MI(X;Y) = \sum_{x \in \mathcal{X}} \sum_{y \in \mathcal{Y}} P_{XY}(x,y) \log \frac{P_{XY}(x,y)}{P_X(x)P_Y(y)},$$

where $P_{XY}(x, y)$ is the joint probability distribution of $X$ and $Y$, and $P_X(x)$ and $P_Y(y)$ are the marginal probability distributions of $X$ and $Y$, respectively. In our context:

- $X$ represents the values of a particular embedding dimension.
- $Y$ represents $S_1$ (0) or $S_2$ (1).

To apply mutual information analysis, we discretize the embedding dimensions using quantile-based binning with 10 bins. This number was selected as a balance between the preservation of information content and the avoidance of excessive complexity in the estimation of the $MI$ score and is a common practice in similar analyses Steuer et al. (2002).

## 4.3 RECURSIVE FEATURE ELIMINATION

We initially examined each embedding dimension's predictive capability with simple logistic regression. Unlike more flexible techniques, logistic regression imposes a linear decision boundary, which was unable to capture the complex patterns defining most linguistic contrasts within the generated embeddings. To capture these relationships, we applied Recursive Feature Elimination (RFE) using `scikit-learn`'s implementation with logistic regression as the base estimator Zeng et al. (2009). Embedding pairs were split into their constituent parts, with `sentence1` embeddings labeled as class 0 and `sentence2` embeddings as class 1, enabling a binary classification setup to highlight dimensions that distinguish the two positions. The RFE procedure iteratively trained a model, assigned importance weights to features, and removed the least important ones until the top 20 features remained.

The dataset was divided into training (80%) and testing (20%) sets with a fixed random seed to ensure consistency. RFE was initialized with a logistic regression classifier (max 1000 iterations), and the selected 20 features were used to train a final logistic regression model. The model's performance was evaluated on the test set using accuracy as the metric.

## 4.4 EDI SCORE CALCULATION

To quantify the contribution of of each embedding dimension to a LP, we introduce the Embedding Dimension Importance (EDI) Score, which is computed for each dimension $d$ and each LP $lp$ as follows:

$$\text{EDI}_{d,lp} = w_1 \cdot -\log p_{d,lp} + w_2 \cdot M_{d,lp} + w_3 \cdot R_{d,lp}$$

where $p_{d,lp}$ is the $p$-value obtained from the Wilcoxon signed-rank test results. $M_{d,lp}$ is the mutual information score. $R_{d,lp}$ is the absolute value of the logistic regression weights after the recursive feature elimination if $d$ remains in the reduced feature set for LP $lp$; otherwise, $R_{d,lp} = 0$. $p_{d,lp}$, $M_{d,lp}$, $R_{d,lp}$ are min-max scaled before the EDI score weighted to calculation to enforce EDI scores to be $\in [0, 1]$. Lastly, $w_1 = 0.6$, $w_2 = 0.2$, and $w_3 = 0.2$. Wilcoxon's test was weighted the most heavily, as it calculates the statistical significance of the differences observed, which our testing showed was a strong predictor of dimension importance.

# 5 EVALUATION

## 5.1 LINGUISTIC PROPERTY CLASSIFIER

To verify the feasibility of using sentence pairs, we calculated embedding difference vectors $D_i = \text{emb}(S_{1i}) - \text{emb}(S_{2i})$ and evaluated them as predictors of LP. To this end, we trained an LP classifier that assigns any given embedding difference vector to one of the tested LPs. The primary goal of this classifier is to assess how well different LPs can be separated in the embedding space. The model was trained using an 80-20 training-test split on the entire LDSP-10 dataset.

## 5.2 EDI SCORE EVALUATION

To systematically assess the effectiveness of EDI scores, we implement a structured evaluation framework consisting of a baseline test and three evaluations experiments. For more details on the algorithms for each evaluation method, refer to Appendix C.

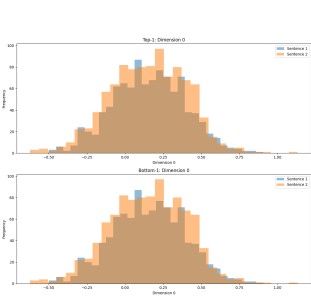
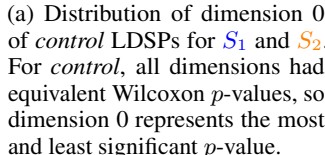
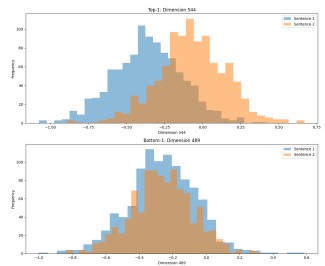
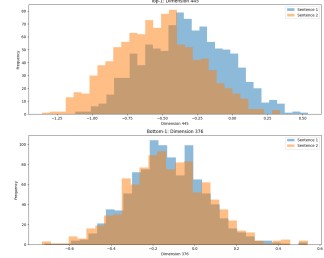

(a) Distribution of dimension 0 of *control* LDSPs for $S_1$ and $S_2$. For *control*, all dimensions had equivalent Wilcoxon $p$-values, so dimension 0 represents the most and least significant $p$-value.

(b) Distribution of dimensions 544 (top) and 489 (bottom), lowest and highest $p$-values respectively, of *negation* LDSPs for $S_1$ and $S_2$. There is a discernible shift to the right in dimension 544, for sentences that are negated.

(c) Distribution of dimensions 445 (top) and 489 (bottom), lowest and highest $p$-values respectively, of *intensifier* LDSPs for $S_1$ and $S_2$. Intensified sentences have values in dimension 445 that tend to be lower, as seen by the distributional shift to the left.

Figure 2: BERT embedding distributions for *control*, *negation*, and *intensifier*.

For the baseline, we train a logistic regression classifier on the full set of embedding dimensions. Given a binary classification task for each LP, the classifier is trained to distinguish between the two sentences in the LDSP using all available embedding dimensions, serving as an upper bound against which subsequent evaluations are compared.

Evaluation 1 explores how dimensions with *high* EDI scores replicate the performance of the full-dimensional classifier. We first rank all dimensions by their EDI score in descending order. Starting with the highest-ranked dimension, we train a logistic regression classifier, as in the baseline evaluation, but only with this single feature. We iteratively add the next highest-ranked dimension, retraining the classifier and evaluating the test accuracy until we reach at least 95% of the baseline accuracy.

Evaluation 2 verifies that dimensions with *low* EDI scores do not encode information relevant to the LP. We identify the 100 lowest-ranked dimensions and train a logistic classifier to distinguish between the two sentences using only those dimensions. We record the accuracy on a test dataset to determine whether it remains close to random chance, as expected, to ensure that these dimensions lack significance in encoding the LP.

Evaluation 3 examines cross-property generalization, exploring whether high-EDI-score dimensions for one LP are specialized rather than broadly informative across different properties. We use the highest-ranked EDI score dimensions of *other* properties to predict the current property. We expect the performance of this classifier to be generally lower than the baseline and the high EDI Score accuracy.

# 6 RESULTS

In this section, we focus on BERT embeddings as a case study for applying our framework. We focus on showing visualizations for *control*, *negation*, and *intensifier*, but all other LPs and related tables/plots can be found in Appendix . The results for GPT-2 and MPNet were similar, and can be reviewed in detail in Appendix E and Appendix F.

## 6.1 CONTROL AND SYNONYM

The *control* LDSPs consists of completely unrelated sentence pairs. As expected, the results show that there are no significant dimensions in BERT embeddings that encode any relationships. Figure 4 illustrates very little agreement the Wilcoxon signed-rank test, RFE, and mutual information. The Wilcoxon test $p$-values show no dimensions with significant differences in their means, as shown in Figure 2a. The maximum EDI score of 0.3683 is the lowest of all other properties. The embeddings

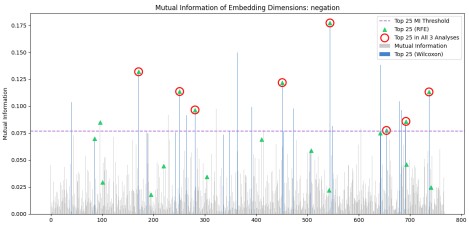 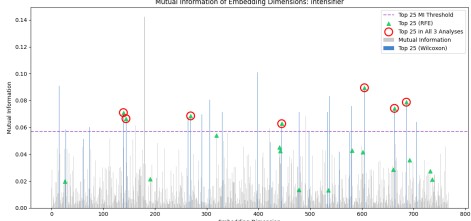

Figure 3: Combined analysis graphs for *negation* and *intensifier*. Circled bars represent dimensions that all three tests agree to be highly important. Similar to Figure 4.

of the two sentences are expected to be far in embedding space because of their unrelated nature, which aligns with these observed results.

Despite having sentences that were very close or equivalent in meaning, the results of the analysis for the *synonym* LDSPs were very close to the completely unrelated sentences of *control*. The Wilcoxon test shows no significant dimensions that encode meaningful differences between the sentences. The maximum EDI score of $0.8751$ is followed by a steep drop-off.

## 6.2 NEGATION AND POLARITY

The *negation* LDSPs showed very strong results, with 13 dimensions with an EDI score of $0.8$ or above. The maximum EDI score of $0.9987$ for dimensions $544$ is one of the strongest out of any LP. Figure 3 illustrates this, with high agreement between the Wilcoxon signed-rank test, RFE, and mutual information test results. Figure 2b highlights the distributional shift in some dimensions, which compared to the *control* highlights a discernible, binary relationship in the data.

*Polarity* is very similar to negation and had similarly strong results. With a maximum EDI score of 0.9977 for dimension 431, and over 20 dimensions with EDI scores over 0.8, it was also one of the strongest relationships that we observed. The singular switch to an antonym in the sentence completely reverses the meaning of the sentence, explaining the strong binary relationship between the sentences.

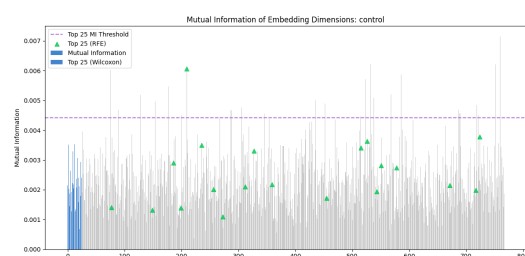

Figure 4: Combined analysis graph for *control*: shows the top 25 important dimensions selected by each of the three methods in § 4. Bar height represents mutual information (MI); bars above the dashed line are in the top 25 MI scores. Blue bars signify the lowest Wilcoxon test $p$-values. Green triangles indicate a dimension that was selected by recursive feature elimination (RFE) with `num_features` set to 25. In the case for *control*, all dimensions had equivalent Wilcoxon $p$-values, so the first 25 are selected.

## 6.3 INTENSIFIER

Adding a word to increase the emphasis of a verb changes the meaning of the sentence to a lesser degree than a complete reversal, so the results of the *intensifier* LDSPs reflect a slightly weaker relationship than *negation*. There are fewer dimensions with multiple test agreement, as shown in Figure 3, as well as a slighter distributional shift, as shown by the most significant $p$-value Wilcoxon test results (Figure 2c). With a maximum EDI score of $0.8911$, the encoding is relatively weaker, but noticeable.

## 6.4 OTHER LINGUISTIC PROPERTIES

Largely syntactical changes, such as those observed in *definiteness*, led to strong EDI scores as well. *Definiteness* had the highest dimensional EDI score, with dimension 180 receiving a score of $1.0$. A

simple switch from a definite to an indefinite article is a distinct change in structure. As articles are present in most English sentences, a singular dimension with a perfect EDI score is expected.

*Voice*, another syntactical property, had pairs of sentences with shuffled word orders and verb changes. The results show that this is encoded in relatively few dimensions, with only 3 dimensions scoring above 0.9.

The *quantity* LDSPs involve changes in the syntax and semantics. Similar to the *intensifier* results, the EDI scores at large were relatively lower for these properties, but still much stronger than the *control*.

*Tense* represented a large semantic change, as well as a structural one in the conjugation of verbs. Although the maximum EDI score of 0.9405 was not as high as other properties, 18 embeddings scored above 0.8, indicating an encoding of this property over many dimensions.

For more details and visualizations of all properties, refer to Appendix D.

## 6.5 EVALUATION RESULTS

The LP classifier achieved a test accuracy of 0.863 with a confusion matrix as shown in Figure 5, demonstrating that the embedding difference vectors contain sufficient separable information to distinguish between different LPs. Moreover, the strong performance of the classifier supports the validity of our pairwise minimal-perturbation approach, indicating that small controlled changes in sentence pairs effectively capture linguistic distinctions in the embedding space.

In the high EDI score evaluation, we observed that across most LPs, only less than 12 of the highest-ranked dimensions were required to recover at least 95% of the baseline classifier's accuracy, with some properties (i.e. *factuality*) requiring as few as four dimensions. This indicates that the information necessary for classifying each LP is concentrated in a relatively small subset of embedding dimensions. Conversely, the low EDI score evaluation confirmed that dimensions with low scores contribute minimally to classification performance. Even when using the 100 lowest-ranked dimensions, the resulting

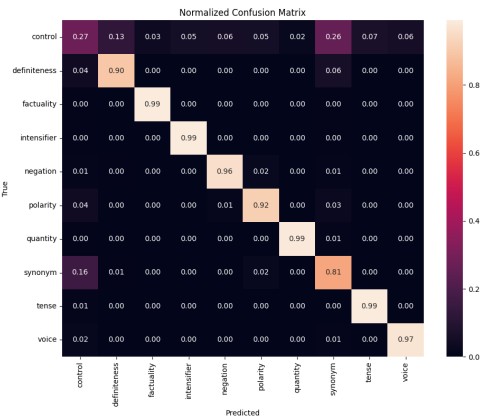

Figure 5: Confusion matrix for the LP classifier (§ 5.1). All LPs, except *control* and *synonym*, are accurately classified by the model. *Control*'s randomness ensures that its different vectors contain no consistent separability, similarly with *synonym*'s unordered pairings.

classifier performed consistently worse than classifiers using much fewer (4-38) of the highest-ranked dimensions (Figures 6a, 6b). This demonstrates the EDI score's validity as a measure of whether a given dimension encodes information relevant to an LP.

Finally, the cross-property evaluation demonstrated that using the top-ranked dimensions from another LP generally resulted in lower classification performance compared to using the high-EDI dimensions of the target property, showing that the EDI score effectively identifies dimensions that encode information specific to each LP. Interestingly, we found that certain properties with conceptual similarities performed best for each other. For example, in the polarity classification task, the top EDI dimensions from negation achieved the highest accuracy among all cross-property evaluations, reaching 0.895 (Figure 6a). This result aligns with the intuition that negative sentiment—typically represented by the second sentence in polarity pairs—is often expressed through negation, reinforcing the semantic connection between these LPs.

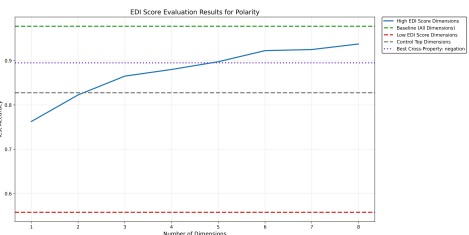
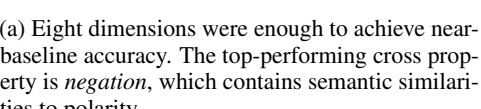
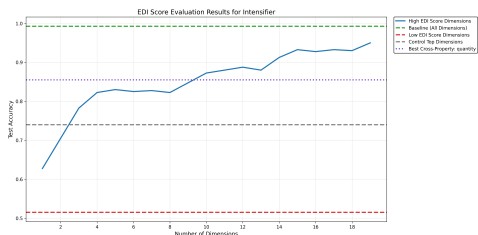

(a) Eight dimensions were enough to achieve near-baseline accuracy. The top-performing cross property is *negation*, which contains semantic similarities to polarity.

(b) Incrementally added 19 high-EDI dimensions until the classifier reached near-baseline performance. Low-EDI performance (red dashed line) was nearly half.

Figure 6: Evaluation plots for *polarity* and *intensifier*. Green dashed line marks the baseline performance threshold, the grey dashed line is the performance of the top EDI dimensions on *control*, and the red dashed line is the performance of the lowest 100 EDI dimensions. The blue line tracks the test accuracy of the classifier as we increased the number of top EDI-scored dimensions.

## 7 DISCUSSION

The results of this study provide a clear demonstration of the ability to disentangle specific LPs within high-dimensional embeddings. Our analysis shows that certain LPs are robustly encoded in distinct embedding dimensions, as evidenced by high Embedding Dimension Importance (EDI) scores and agreement across multiple analytical methods. These methods were chosen after rigorous experimentation, where principal component analysis, simple logistic regression, and other methods were rejected due to their inability to capture the nuanced, non-linear information encoded in these embeddings. Negation yielded one of the the highest maximum EDI scores and a significant number of dimensions with high interpretability. This supports the notion that negation is a well-structured and salient linguistic feature in BERT embeddings.

In contrast, some properties exhibited minimal evidence of dimension-specific encoding, which we hypothesize to be due to a lack of a binary or clear-cut way of encoding these relationships. Synonymy showed low maximum EDI scores and inconsistent results across the Wilcoxon Signed-Rank Test, Mutual Information, and Recursive Feature Elimination. Synonym pairs in our dataset could be permuted without affecting the consistency of the data, and 0-1 labels for our classifiers and mutual information were meaningless; therefore, our methods are unable to extract the dimensional distribution of synonym encodings.

In summary, this study underscores the heterogeneous nature of linguistic encoding in BERT embeddings, with some properties exhibiting clear, interpretable patterns while others remain elusive. The proposed EDI score and analytical framework provide valuable tools for advancing the interpretability of embeddings, with implications for bias mitigation, model optimization, and the broader goal of responsible AI deployment.

## 8 LIMITATIONS

While our study provides insight into the interpretability of embedding dimensions, it is constrained primarily due to data availability. Generating high-quality LDSPs with LLM-based tools is difficult, as ensuring diversity, minimal redundancy, and high linguistic quality becomes significantly more difficult with more data generated. Overly simplistic, repetitive outputs are difficult to avoid, despite careful prompt engineering.

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

## A  DATASET GENERATION PIPELINE

Figure 7 illustrates the procedure used to generate the LDSP-10 dataset. The batch procedure of generating 100 pairs of sentences at a time was crucial in minimizing API costs while also getting high-quality generations that would be useful for our experiments. The prompt template used can be seen in Figure 8.

## B  LINGUISTIC PROPERTY DEFINITIONS

We tested LDSPs for the following linguistic properties:

- *Definiteness* involves the use of definite or indefinite articles within a sentence, such as *the* compared to *a*, respectively.
- *Factuality* refers to the degree of truth implied by the structure of the sentence.
- *Intensifier* refers to the degree of emphasis present within a sentence.
- *Negation* occurs when a *not* is added to a sentence, negating the meaning.
- *Polarity* this is similar to a negation, and occurs when an antonym is added, reversing the meaning of the sentence completely.
- *Quantity* a switch from an exact number used to numerate the items to a grouping word.
- *Synonym* both sentences have the same meaning, with one word being replaced by one of its synonyms.
- *Tense* one sentence is constructed in the present tense, while the other is in the past tense.

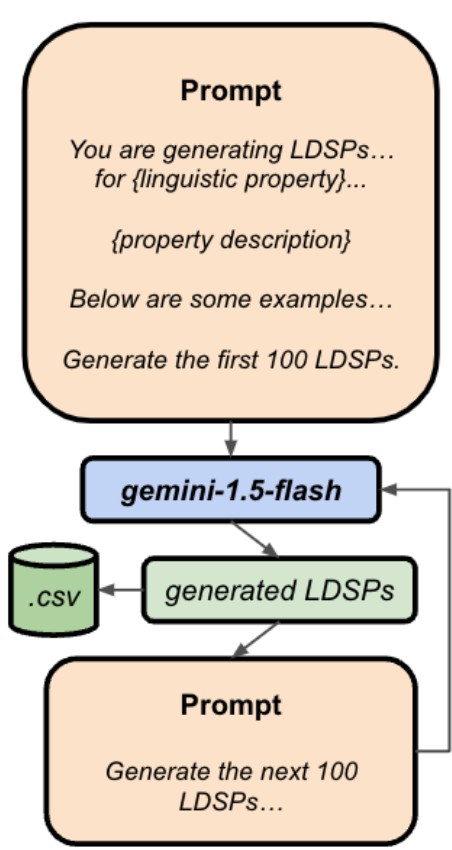

Figure 7: LDSP generation pipeline with Google's `gemini-1.5-flash` model API.

```
prompt_template = """

You are generating a dataset of Linguistically Distinct Sentence Pairs (LDSPs).
Each LDSP will differ in one key linguistic property while maintaining the same
 overall meaning.

Below are some examples of LDSPs

Linguistic Property: negation
LDSP: ('The box is on the counter', 'The box is not on the counter')

Linguistic Property: tense
LDSP: ('The box is on the counter', 'The box was on the counter')

You will generate {num_ldsps} distinct LDSPs of various topics, 100 at a time.

You will generate them as two columns of a CSV. One column for first sentence of
 the LDSP, and the other column for the second.
Each row is a new LDSP, so you will generate {num_ldsps} rows in total.

Generate no other text. Vary the sentence structure.

The property for which you will be generating LDSPs will be {linguistic_property}.

Property Description: {property_description}

An example LDSP for this property is
{example_ldsp}

Generate the first 100 LDSPs.

"""
```

Figure 8: The prompt template used to generate LDSPs with the gemini-1.5-flash model API.

## C    EVALUATION ALGORITHMS

To systematically assess the efficacy of EDI (Embedding Dimension Importance) scores, we conduct a structured evaluation using logistic regression classifiers. Our evaluation consists of three key evaluation algorithms:

---

**Algorithm 1** Evaluation 1: High EDI Score

---

**Require:** Ranked dimensions $D = \{d_1, d_2, ..., d_{768}\}$ sorted by descending EDI score
**Ensure:** Accuracy curve $A_k$ as a function of dimensions used
  1: Initialize $k \leftarrow 1$, $A_k \leftarrow 0$
  2: **while** $A_k < 0.95 A_{\text{baseline}}$ **do**
  3:     Select top $k$ dimensions: $X_k = X[:, D_{1:k}]$
  4:     Train logistic regression on $X_k$
  5:     Compute test accuracy $A_k \leftarrow \text{Evaluate}(\theta, X_{\text{test}}, y_{\text{test}})$
  6:     $k \leftarrow k + 1$
  7: **end while**
  8: **return** $A_k$

---

**Algorithm 2** Evaluation 2: Low EDI Score

---

**Require:** Ranked dimensions $D = \{d_1, d_2, ..., d_{768}\}$ sorted by ascending EDI score
**Ensure:** Test accuracy $A_{\text{low}}$ using lowest-EDI dimensions
  1: Select bottom $k = 100$ dimensions: $X_{\text{low}} = X[:, D_{1:100}]$
  2: Train logistic regression on $X_{\text{low}}$
  3: Compute test accuracy $A_{\text{low}} \leftarrow \text{Evaluate}(\theta, X_{\text{test}}, y_{\text{test}})$
  4: **return** $A_{\text{low}}$

---

**Algorithm 3** Evaluation 3: Cross-Property

---

**Require:** Current property $P_0$ dataset $(X, y)$, set of other properties $\mathcal{P} = \{P_1, P_2, ..., P_9\}$, where each $P_i$ has ranked EDI dimensions $D_{P_i}$
**Ensure:** Accuracy scores $\{A_{P_1}, A_{P_2}, ..., A_{P_9}\}$
  1: **for** each property $P \in \mathcal{P}$ **do**
  2:     Retrieve top $k = 25$ dimensions from $P$: $D_P^{1:25}$
  3:     Extract these dimensions from current data: $X_{\text{train}}^P = X_{\text{train}}[:, D_P^{1:25}]$
  4:     Train logistic regression on $X_{\text{train}}^P$
  5:     Compute test accuracy $A_P \leftarrow \text{Evaluate}(\theta, X_{\text{test}}^P, y_{\text{test}})$
  6: **end for**
  7: **return** $\{A_P\}_{P \in \mathcal{P}}$

---

These evaluations provide a comprehensive understanding of how EDI scores relate to classification accuracy, ensuring that high EDI dimensions contain useful linguistic information while low EDI dimensions do not. The cross-property evaluation further confirms that high-EDI dimensions are specialized rather than general indicators of LPs.

## D    ADDITIONAL LINGUISTIC PROPERTY RESULTS FOR BERT EMBEDDINGS

### D.1    CONTROL

Table 3 highlights the top 10 EDI scores for the *control*. The baseline evaluation results for *control* showed an accuracy of 0.5200, close to random chance. The Low EDI score test yielded an accuracy of 0.4575. The High EDI score test demonstrated quick improvements, achieving 95% of baseline accuracy with a single dimension, as the baseline accuracy was low, as illustrated in Figure 9. The greatest cross-property accuracy was achieved by *voice*, at 0.5325.

| Dimension | EDI Score |
|-----------|-----------|
| 209 | 0.3683 |
| 526 | 0.2639 |
| 578 | 0.2434 |
| 235 | 0.2342 |
| 186 | 0.2315 |
| 515 | 0.2196 |
| 724 | 0.2167 |
| 760 | 0.2000 |
| 327 | 0.1958 |
| 551 | 0.1913 |

Table 3: Top 10 BERT EDI scores for the *Control*.

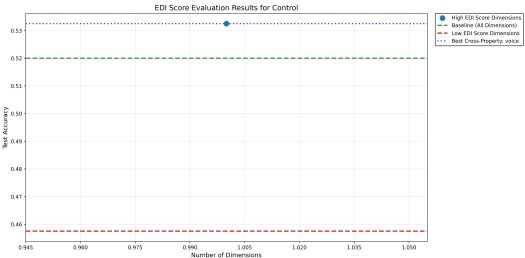

Figure 9: Evaluation plot for *control*. The blue dot indicates that with just 1 high-EDI dimension, the classifier was able to achieve performance better than the baseline. However, in the case of *control*, all the accuracies are near 0.5 (random-choice accuracy), as expected.

## D.2 DEFINITENESS

Definiteness had some of the strongest results out of any LP. Figure 10 highlight the difference between the most prominent dimensions for this property. Table 4 highlights the top 10 EDI scores, while Figure 12 illustrates the high level of agreement between our various tests.

The baseline evaluation results for *definiteness* showed an accuracy of 0.9450. The Low EDI score test yielded an accuracy of 0.5425, very close to random chance. The High EDI score test was able to achieve 95% of baseline accuracy with 25 dimensions, as illustrated in Figure 11. The greatest cross-property accuracy was achieved by intensifier, at 0.8425.

## D.3 FACTUALITY

Factuality had strong results. Figure 13 highlights the stark difference between the most prominent dimensions encoding this property. Table 5 highlights the top 10 EDI scores, while Figure 15 illustrates the high level of agreement between our various tests.

| Dimension | EDI Score |
|-----------|-----------|
| 180 | 1.0000 |
| 123 | 0.8824 |
| 319 | 0.8819 |
| 385 | 0.8639 |
| 109 | 0.8155 |
| 497 | 0.7974 |
| 683 | 0.7948 |
| 172 | 0.7926 |
| 430 | 0.7907 |
| 286 | 0.7862 |

Table 4: Top 10 BERT EDI scores for *Definiteness*.

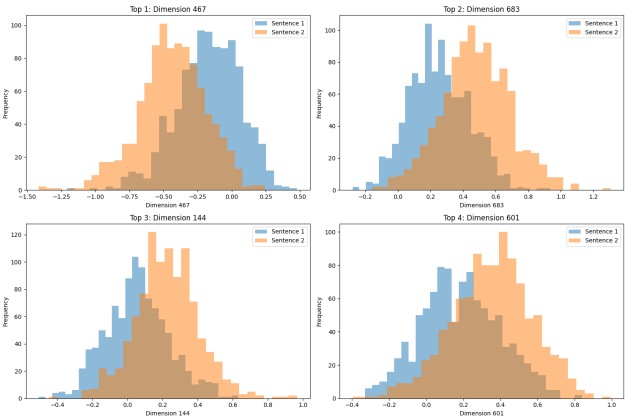

Figure 10: BERT Dimensional Embedding values for the Wilcoxon test results with the most significant p-values for *Definiteness*.

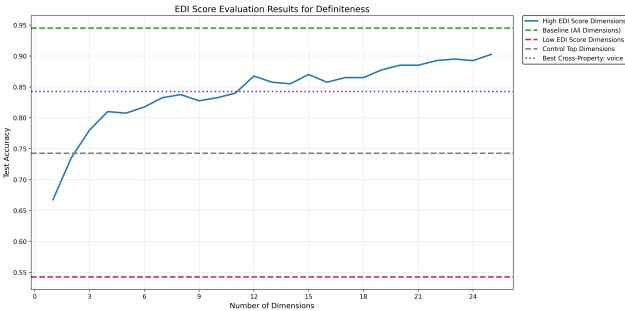

Figure 11: High EDI score evaluation results for BERT Embeddings of *definiteness*.

The baseline evaluation results for *factuality* showed an accuracy of 0.9975. The Low EDI score test yielded an accuracy of 0.5975, approximately random. The High EDI score test demonstrated very quick improvements, achieving 95% of baseline accuracy with 4 dimensions, as illustrated in Figure 14. The greatest cross-property accuracy was achieved by *tense*, at 0.9650.

## D.4 INTENSIFIER

Table 6 highlights the top 10 EDI scores for *intensifier*. The baseline evaluation results for *intensifier* showed an accuracy of 0.9925. The Low EDI score test yielded an accuracy of 0.5150, close to random chance. The High EDI score test demonstrated incremental improvements, achieving 95%

| Dimension | EDI Score |
|-----------|-----------|
| 577 | 0.9740 |
| 43 | 0.9386 |
| 210 | 0.9249 |
| 745 | 0.8954 |
| 539 | 0.8887 |
| 387 | 0.8869 |
| 60 | 0.8727 |
| 16 | 0.8617 |
| 54 | 0.8609 |
| 97 | 0.8538 |

Table 5: Top 10 BERT EDI scores for *Factuality*.

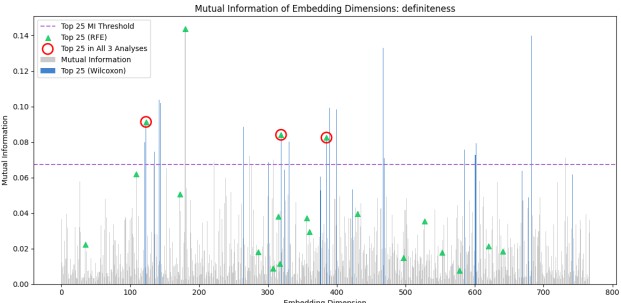

Figure 12: BERT Mutual Information of Embedding Dimensions overlaid with Wilcoxon test and RFE results for *Definiteness*

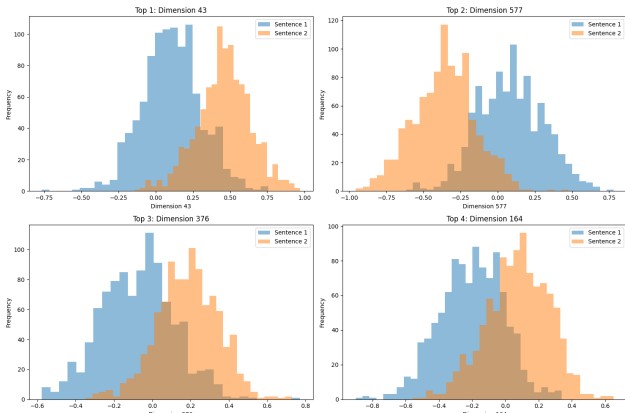

Figure 13: BERT Dimensional Embedding values for the Wilcoxon test results with the most significant p-values for *Factuality*.

of baseline accuracy with 19 dimensions, as illustrated in Figure 6b. The greatest cross-property accuracy was achieved by *quantity*, at 0.8550.

### D.5 NEGATION

Table 7 highlights the top 10 EDI scores for *negation*. The baseline evaluation results for *negation* showed an accuracy of 0.9925. The Low EDI score test yielded an accuracy of 0.5800, close to random chance. The High EDI score test demonstrated incremental improvements, achieving 95% of baseline accuracy with 11 dimensions, as illustrated in Figure 16. The greatest cross-property accuracy was achieved by *tense*, at 0.9100.

| Dimension | EDI Score |
|-----------|-----------|
| 686 | 0.8911 |
| 663 | 0.8832 |
| 139 | 0.8805 |
| 605 | 0.8790 |
| 269 | 0.8650 |
| 441 | 0.8612 |
| 144 | 0.8535 |
| 692 | 0.8468 |
| 445 | 0.8385 |
| 442 | 0.8221 |

Table 6: Top 10 BERT EDI scores for *Intensifier*.

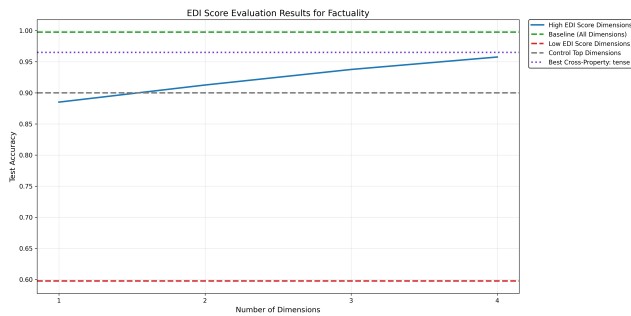

Figure 14: High EDI score evaluation results for BERT Embeddings of *factuality*.

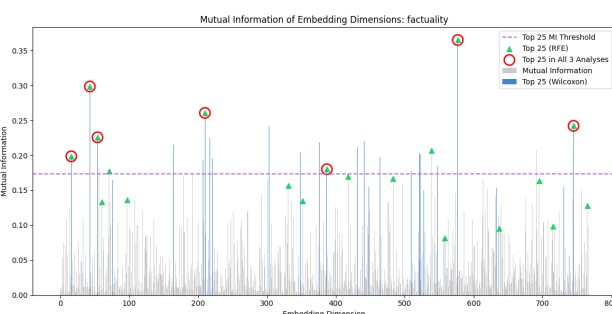

Figure 15: Mutual Information of Embedding Dimensions overlaid with Wilcoxon test and RFE results for *Factuality*

### D.6 POLARITY

Polarity, as it is similar to negation, had extremely strong results. Figure 17 highlights the differences between the most prominent dimensions encoding this property. Table 8 highlights the top 10 EDI scores, while Figure 18 illustrates the extremely high level of agreement between our various tests.

The baseline evaluation results for *polarity* showed an accuracy of 0.9775. The Low EDI score test yielded an accuracy of 0.5575, close to random chance. The High EDI score test demonstrated incremental improvements, achieving 95% of baseline accuracy with 8 dimensions, as illustrated in Figure 6a. The greatest cross-property accuracy was achieved by *negation*, at 0.8950.

### D.7 QUANTITY

*Quantity* had more moderate results compared to *polarity* and *negation*. Figure 19 highlights the difference between the most prominent dimensions encoding this property. Table 9 highlights the top 10 EDI scores, while Figure 21 illustrates the moderate level of agreement the tests.

| Dimension | EDI Score |
|-----------|-----------|
| 544 | 0.9987 |
| 251 | 0.9277 |
| 171 | 0.9236 |
| 451 | 0.9101 |
| 737 | 0.8891 |
| 281 | 0.8812 |
| 96 | 0.8624 |
| 692 | 0.8512 |
| 85 | 0.8501 |
| 642 | 0.8461 |

Table 7: Top 10 BERT EDI scores for *Negation*.

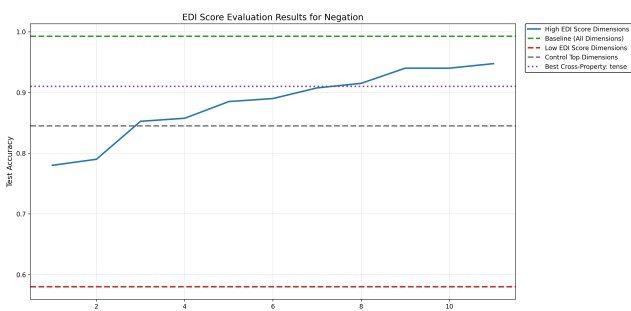

Figure 16: High EDI score evaluation results for BERT Embeddings of *Negation*.

| Dimension | EDI Score |
|-----------|-----------|
| 431 | 0.9947 |
| 623 | 0.9867 |
| 500 | 0.9675 |
| 461 | 0.9200 |
| 96 | 0.9063 |
| 505 | 0.8910 |
| 594 | 0.8745 |
| 407 | 0.8492 |
| 397 | 0.8459 |
| 613 | 0.8445 |

Table 8: Top 10 BERT EDI scores for *Polarity*.

The baseline evaluation results for *quantity* showed an accuracy of 1.0000. The Low EDI score test yielded an accuracy of 0.6425. The High EDI score test demonstrated incremental improvements, achieving 95% of baseline accuracy with 9 dimensions, as illustrated in Figure 20. The greatest cross-property accuracy was achieved by *intensifier*, at 0.9025.

## D.8 SYNONYM

Table 10 highlights the top 10 EDI scores for *synonym*. Figure 22 highlights the differences between the most prominent dimensions that encode this property.

The baseline evaluation results for *synonym* showed an accuracy of 0.7400. The Low EDI score test yielded an accuracy of 0.4625, slightly above random chance. The High EDI score test demonstrated very slow improvements, achieving 95% of baseline accuracy with 392 dimensions, as illustrated in Figure 23. The greatest cross-property accuracy was achieved by *quantity*, at 0.6175.

| Dimension | EDI Score |
|-----------|-----------|
| 463 | 0.9316 |
| 457 | 0.9155 |
| 390 | 0.9050 |
| 243 | 0.8866 |
| 192 | 0.8777 |
| 735 | 0.8545 |
| 489 | 0.8525 |
| 67 | 0.8430 |
| 304 | 0.8384 |
| 723 | 0.8217 |

Table 9: Top 10 BERT EDI scores for *Quantity*.

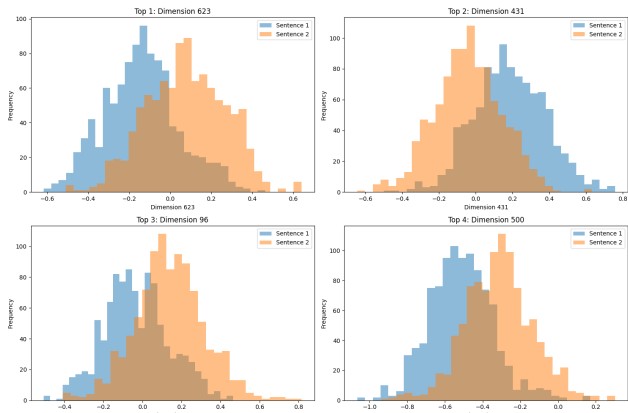

Figure 17: BERT Dimensional Embedding values for the Wilcoxon test results with the most significant p-values for *Polarity*.

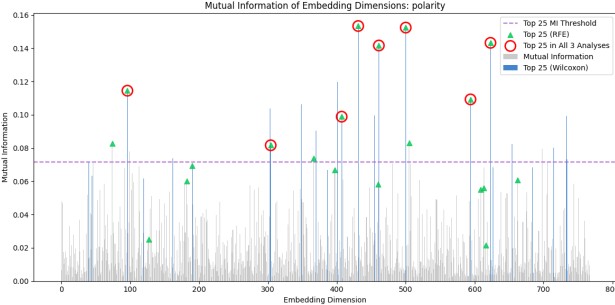

Figure 18: Mutual Information of BERT Embedding Dimensions overlaid with Wilcoxon test and RFE results for *Polarity*

## D.9 TENSE

*Tense* had moderate results. Figure 24 highlights the differences between the most prominent dimensions encoding this property. Table 11 highlights the top 10 EDI scores, while Figure 27 illustrates the level of agreement the tests.

The baseline evaluation results for *tense* showed an accuracy of 0.9975. The Low EDI score test yielded an accuracy of 0.4625, close to random chance. The High EDI score test demonstrated incremental improvements, achieving 95% of baseline accuracy with 11 dimensions, as illustrated in Figure 25. The greatest cross-property accuracy was achieved by *control*, at 0.9150.

| Dimension | EDI Score |
|-----------|-----------|
| 676 | 0.8751 |
| 203 | 0.7744 |
| 701 | 0.6916 |
| 654 | 0.6897 |
| 463 | 0.6889 |
| 544 | 0.6602 |
| 91 | 0.6598 |
| 437 | 0.6557 |
| 446 | 0.6543 |
| 487 | 0.6415 |

Table 10: Top 10 BERT EDI scores for *Synonym*.

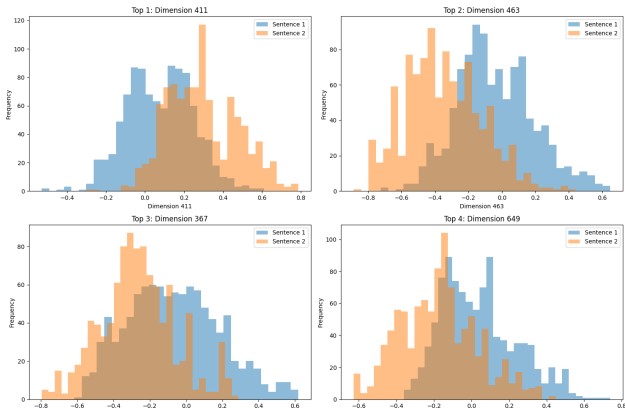

Figure 19: BERT Dimensional Embedding values for the Wilcoxon test results with the most significant p-values for *Quantity*.

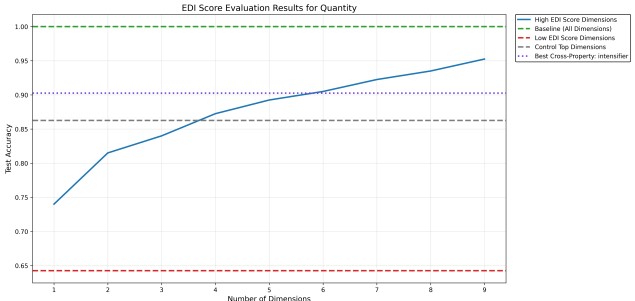

Figure 20: High EDI score evaluation results for BERT Embeddings of *quantity*.

### D.10 VOICE

*Voice* had relatively few dimensions with very high EDI scores. Figure 26 highlights the differences between the most prominent dimensions encoding this property. Table 12 highlights the top 10 EDI scores, while Figure 29 illustrates the level of agreement the tests.

The baseline evaluation results for *voice* showed an accuracy of 1.0000. The Low EDI score test yielded an accuracy of 0.5200, close to random chance. The High EDI score test demonstrated incremental improvements, achieving 95% of baseline accuracy with 30 dimensions, as illustrated in Figure 28. The greatest cross-property accuracy was achieved by *definiteness*, at 0.8400.

| Dimension | EDI Score |
|-----------|-----------|
| 641 | 0.9405 |
| 586 | 0.9369 |
| 335 | 0.9162 |
| 38 | 0.9113 |
| 684 | 0.8977 |
| 522 | 0.8908 |
| 470 | 0.8880 |
| 548 | 0.8821 |
| 4 | 0.8812 |
| 653 | 0.8627 |

Table 11: Top 10 BERT EDI scores for *Tense*.

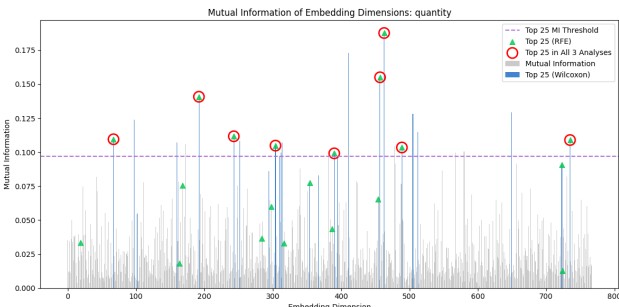

Figure 21: Mutual Information of BERT Embedding Dimensions overlaid with Wilcoxon test and RFE results for *Quantity*

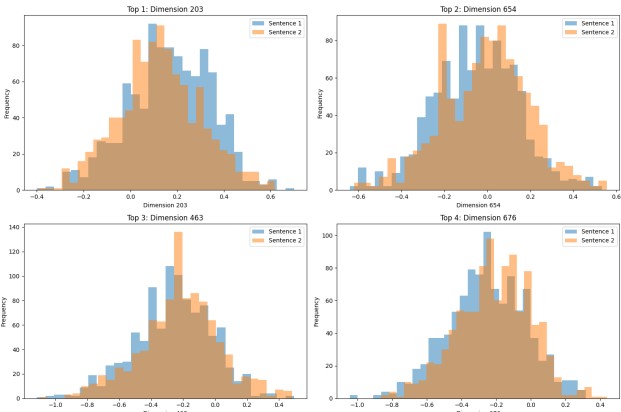

Figure 22: BERT Dimensional Embedding values for the Wilcoxon test results with the most significant p-values for *Synonym*.

# E    GPT-2

This section will contain the visualizations of the results for GPT-2 embeddings. Full detailed results, including full EDI scores as well as additional visualization, will be available on GitHub upon publication.

## E.1    LINGUISTIC PROPERTY CLASSIFIER

The results from the Linguistic Property Classifier for GPT-2 embeddings is shown in Figure 30.

| Dimension | EDI Score |
|-----------|-----------|
| 653 | 0.9722 |
| 523 | 0.9552 |
| 766 | 0.9376 |
| 27 | 0.8875 |
| 111 | 0.8783 |
| 286 | 0.8586 |
| 222 | 0.8437 |
| 693 | 0.8404 |
| 16 | 0.8182 |
| 95 | 0.8113 |

Table 12: Top 10 BERT EDI scores for *Voice*.

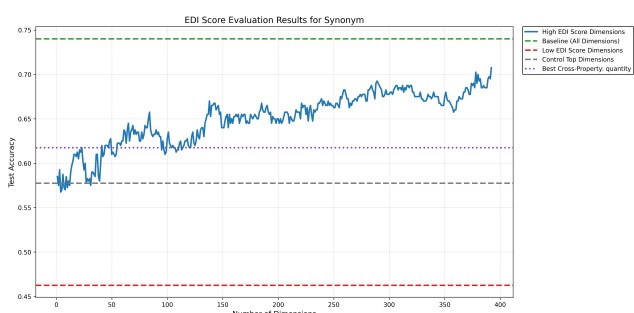

Figure 23: High EDI score evaluation results for BERT Embeddings of *synonym*.

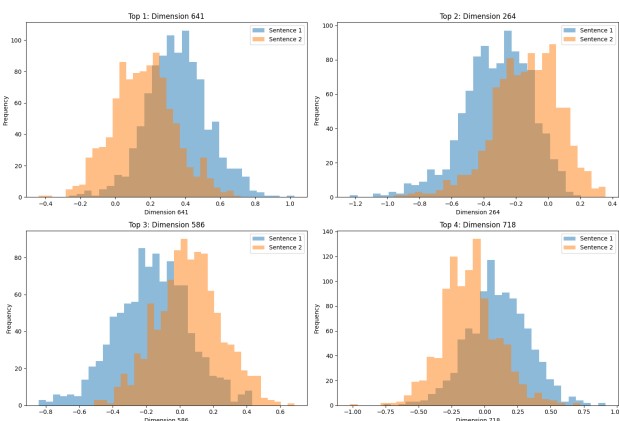

Figure 24: BERT Dimensional Embedding values for the Wilcoxon test results with the most significant p-values for *Tense*.

### E.2 CONTROL

Figure 31 highlights the difference between the most prominent dimensions encoding this property. Figure 33 illustrates the level of agreement between the tests.

The baseline evaluation results for *control* showed an accuracy of 0.4725, close to chance. The Low EDI score test yielded an accuracy of 0.4400. The High EDI score test demonstrated strong performance, achieving 95% of baseline accuracy with just a single dimension, as the baseline accuracy was close to random chance, as illustrated in Figure 32. The highest cross-property accuracy was achieved by *voice*, at 0.5450.

### E.3 DEFINITENESS

Figure 34 highlights the difference between the most prominent dimensions encoding this property. Figure 36 illustrates the level of agreement between the tests.

The baseline evaluation results for *definiteness* showed an accuracy of 0.9575. The Low EDI score test yielded an accuracy of 0.5000. The High EDI score test demonstrated strong performance, achieving 95% of baseline accuracy with just a single dimension, as illustrated in Figure 35. The highest cross-property accuracy was achieved by *intensifier*, at 0.9400, followed closely by *factuality* (0.9325) and *synonym* (0.9275).

### E.4 FACTUALITY

Figure 37 highlights the difference between the most prominent dimensions encoding this property. Figure 39 illustrates the level of agreement between the tests.

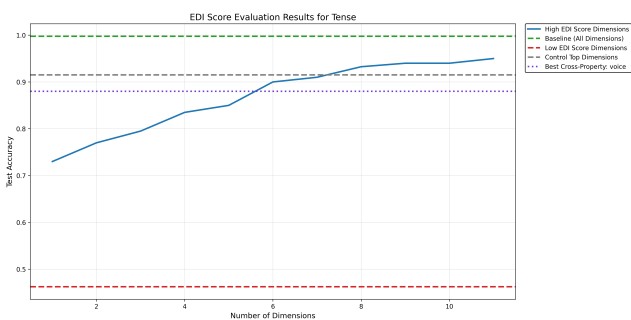

Figure 25: High EDI score evaluation results for BERT Embeddings of *tense*.

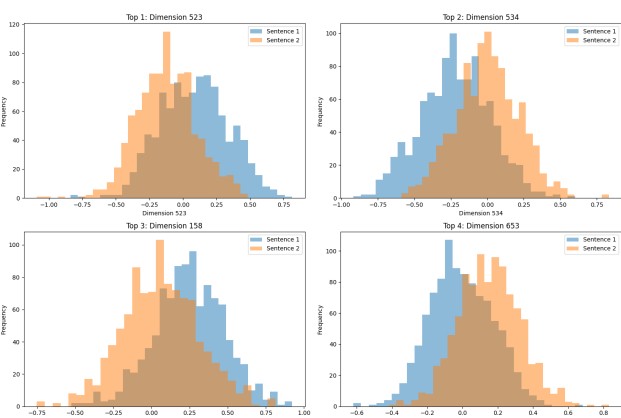

Figure 26: BERT Dimensional Embedding values for the Wilcoxon test results with the most significant p-values for *Voice*.

The baseline evaluation results for *factuality* showed an accuracy of 1.0000. The Low EDI score test yielded an accuracy of 0.6800. The High EDI score test demonstrated strong performance, achieving 95% of baseline accuracy with just a single dimension, as illustrated in Figure 38. The highest cross-property accuracy was achieved by *negation*, at 0.9975.

### E.5  INTENSIFIER

Figure 40 highlights the difference between the most prominent dimensions encoding this property. Figure 42 illustrates the level of agreement between the tests.

The baseline evaluation results for *intensifier* showed an accuracy of 1.0000. The Low EDI score test yielded an accuracy of 0.5825. The High EDI score test demonstrated steady improvement, reaching 95% of baseline accuracy with 4 dimensions, as illustrated in Figure 41. The highest cross-property accuracy was achieved by *definiteness*, at 0.9600.

### E.6  NEGATION

Figure 43 highlights the difference between the most prominent dimensions encoding this property. Figure 45 illustrates the level of agreement between the tests.

The baseline evaluation results for *negation* showed an accuracy of 0.9850. The Low EDI score test yielded an accuracy of 0.5450. The High EDI score test demonstrated steady improvement, reaching 95% of baseline accuracy with 6 dimensions, as illustrated in Figure 44. The highest cross-property accuracy was achieved by *intensifier*, at 0.9475.

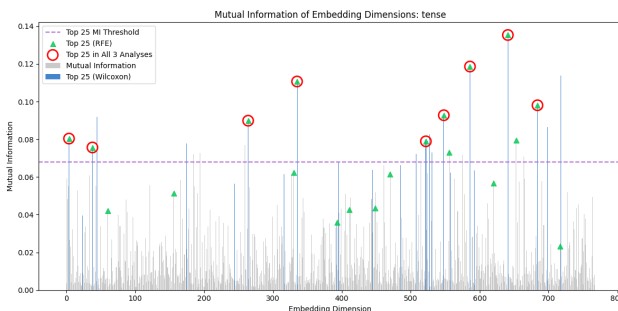

Figure 27: Mutual Information of BERT Embedding Dimensions overlaid with Wilcoxon test and RFE results for *Tense*

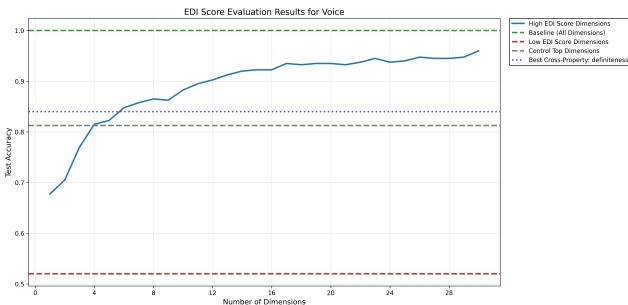

Figure 28: High EDI score evaluation results for BERT Embeddings of *voice*.

### E.7 POLARITY

Figure 46 highlights the difference between the most prominent dimensions encoding this property. Figure 48 illustrates the level of agreement between the tests.

The baseline evaluation results for *polarity* showed an accuracy of 0.9975. The Low EDI score test yielded an accuracy of 0.4700. The High EDI score test demonstrated slow improvement, reaching 95% of baseline accuracy with 28 dimensions, as illustrated in Figure 47. The highest cross-property accuracy was achieved by *quantity*, at 0.8300.

### E.8 QUANTITY

Figure 49 highlights the difference between the most prominent dimensions encoding this property. Figure 51 illustrates the level of agreement between the tests.

The baseline evaluation results for *quantity* showed an accuracy of 0.9975. The Low EDI score test yielded an accuracy of 0.6875. The High EDI score test demonstrated steady improvement, reaching 95% of baseline accuracy with 8 dimensions, as illustrated in Figure 50. The highest cross-property accuracy was achieved by *polarity*, at 0.9300.

### E.9 SYNONYM

Figure 52 highlights the difference between the most prominent dimensions encoding this property. Figure 54 illustrates the level of agreement between the tests.

The baseline evaluation results for *synonym* showed an accuracy of 0.6300. The Low EDI score test yielded an accuracy of 0.3575. The High EDI score test demonstrated gradual improvement, reaching 95% of baseline accuracy with 26 dimensions, as illustrated in Figure 53. The highest cross-property accuracy was achieved by *intensifier* at 0.5350.

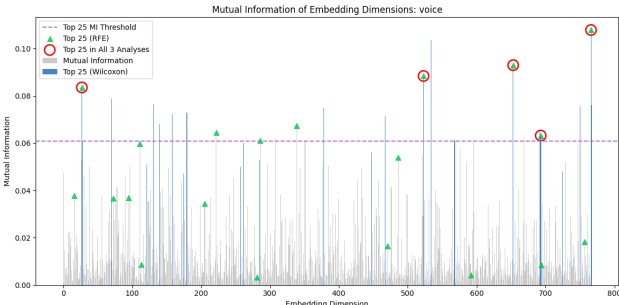

Figure 29: Mutual Information of BERT Embedding Dimensions overlaid with Wilcoxon test and RFE results for *Voice*

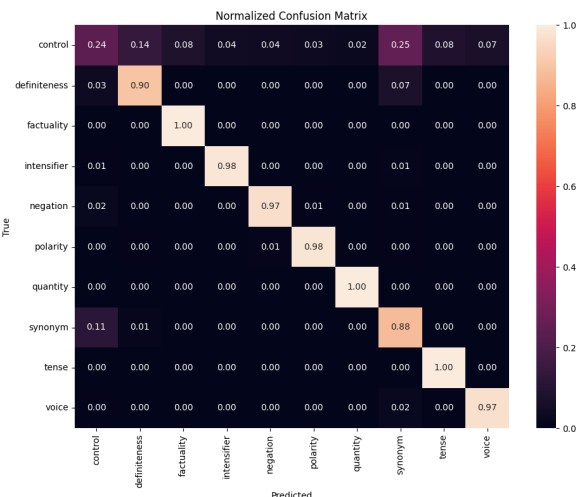

Figure 30: Linguistic Property Classifier results for GPT-2.

### E.10 TENSE

Figure 55 highlights the difference between the most prominent dimensions encoding this property. Figure 57 illustrates the level of agreement between the tests.

The baseline evaluation results for *tense* showed an accuracy of 0.9950. The Low EDI score test yielded an accuracy of 0.4500. The High EDI score test demonstrated slow improvement, reaching 95% of baseline accuracy with 76 dimensions, as illustrated in Figure 56. The highest cross-property accuracy was observed with *definiteness* at 0.7525.

### E.11 VOICE

Figure 58 highlights the difference between the most prominent dimensions encoding this property. Figure 60 illustrates the level of agreement between the tests.

The baseline evaluation results for *voice* showed an accuracy of 1.0000. The Low EDI score test yielded an accuracy of 0.5325, around random chance. The High EDI score test demonstrated significant improvement, reaching 95% of baseline accuracy with just a single dimension, as illustrated in Figure 59. The highest cross-property accuracy was observed with *intensifier* at 0.9900.

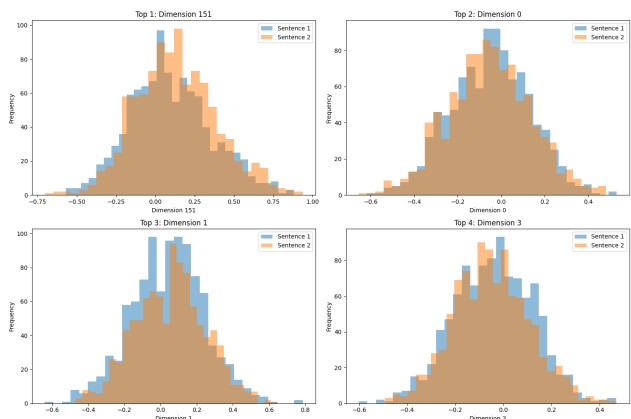

Figure 31: GPT-2 Dimensional Embedding values for the Wilcoxon test results with the most significant p-values for *Control*.

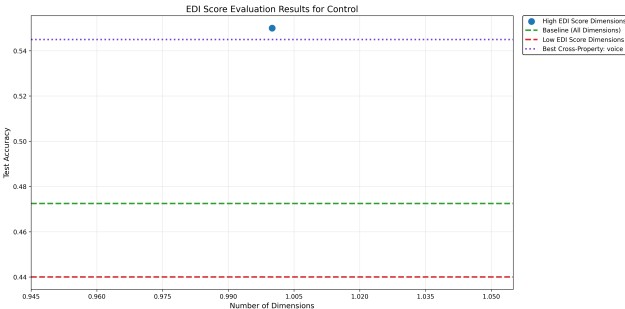

Figure 32: High EDI score evaluation results for GPT-2 Embeddings of *Control*.

# F    MPNET

This section will contain the visualizations of the results for MPNet embeddings. Full detailed results, including full EDI scores as well as additional visualization, will be available on GitHub upon publication.

## F.1    LINGUISTIC PROPERTY CLASSIFIER

The results from the Linguistic Property Classifier for MPNet embeddings is shown in Figure 61.

## F.2    CONTROL

Figure 62 highlights the difference between the most prominent dimensions encoding this property. Figure 64 illustrates the level of agreement between the tests.

The baseline evaluation results for *control* showed an accuracy of 0.4800, which is close to random chance. The Low EDI score test yielded an accuracy of 0.4125. The High EDI score test demonstrated weak performance, achieving 95% of baseline accuracy with just a single dimension, but that is because the baseline accuracy was super close to chance, as illustrated in Figure 63. The highest cross-property accuracy was achieved by *tense*, at 0.5175.

## F.3    DEFINITENESS

Figure 65 highlights the difference between the most prominent dimensions encoding this property. Figure 67 illustrates the level of agreement between the tests.

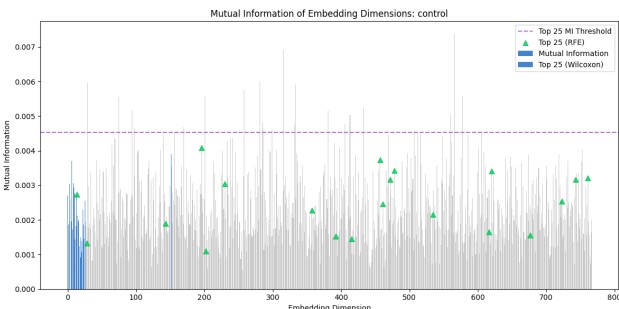

Figure 33: Mutual Information of GPT-2 Embedding Dimensions overlaid with Wilcoxon test and RFE results for *Control*.

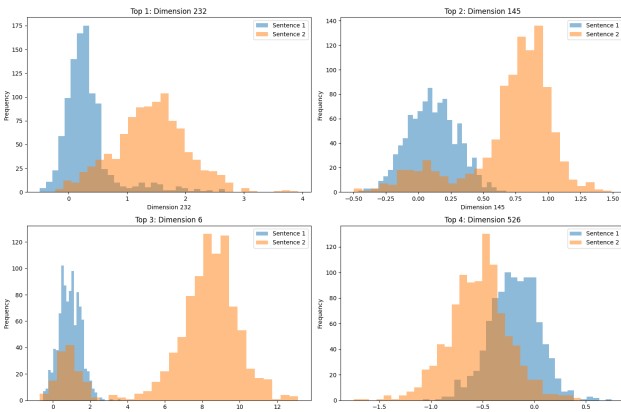

Figure 34: GPT-2 Dimensional Embedding values for the Wilcoxon test results with the most significant p-values for *Definiteness*.

The baseline evaluation results for *definiteness* showed an accuracy of 0.9000. The Low EDI score test yielded an accuracy of 0.4000. The High EDI score test demonstrated strong performance, achieving 95% of baseline accuracy with just a single dimension, as illustrated in Figure 66. The highest cross-property accuracy was achieved by *intensifier*, at 0.6750.

### F.4 FACTUALITY

Figure 68 highlights the difference between the most prominent dimensions encoding this property. Figure 70 illustrates the level of agreement between the tests.

The baseline evaluation results for *factuality* showed an accuracy of 0.9975. The Low EDI score test yielded an accuracy of 0.4825. The High EDI score test demonstrated steady performance, achieving 95% of baseline accuracy with 16 dimensions, as illustrated in Figure 69. The highest cross-property accuracy was achieved by *quantity*, at 0.8875.

### F.5 INTENSIFIER

Figure 71 highlights the difference between the most prominent dimensions encoding this property. Figure 73 illustrates the level of agreement between the tests.

The baseline evaluation results for *intensifier* showed an accuracy of 0.9000. The Low EDI score test yielded an accuracy of 0.4200. The High EDI score test demonstrated slow performance, achieving 95% of baseline accuracy with 347 dimensions, as illustrated in Figure 72. The highest cross-property accuracy was achieved by *quantity*, at 0.6825.

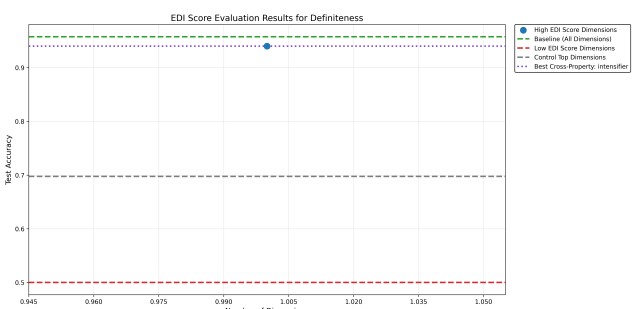

Figure 35: High EDI score evaluation results for GPT-2 Embeddings of *Definiteness*.

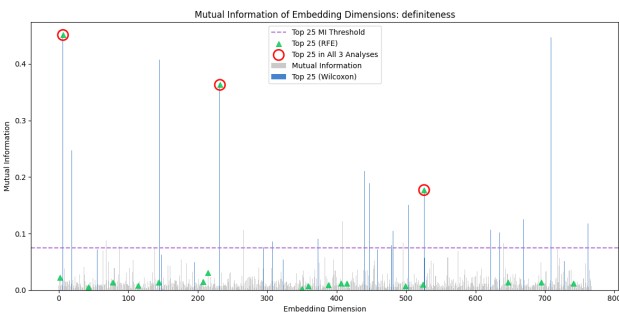

Figure 36: Mutual Information of GPT-2 Embedding Dimensions overlaid with Wilcoxon test and RFE results for *Definiteness*.

## F.6 NEGATION

Figure 74 highlights the difference between the most prominent dimensions encoding this property. Figure 76 illustrates the level of agreement between the tests.

The baseline evaluation results for *negation* showed an accuracy of 0.9750. The Low EDI score test yielded an accuracy of 0.6025. The High EDI score test demonstrated steady improvement, reaching 95% of baseline accuracy with 26 dimensions, as illustrated in Figure 75. The highest cross-property accuracy was achieved by *factuality*, at 0.8900.

## F.7 POLARITY

Figure 77 highlights the difference between the most prominent dimensions encoding this property. Figure 79 illustrates the level of agreement between the tests.

The baseline evaluation results for *polarity* showed an accuracy of 0.9850. The Low EDI score test yielded an accuracy of 0.6900. The High EDI score test demonstrated fast improvement, reaching 95% of baseline accuracy with 6 dimensions, as illustrated in Figure 78. The highest cross-property accuracy was achieved by *negation*, at 0.9575.

## F.8 QUANTITY

Figure 80 highlights the difference between the most prominent dimensions encoding this property. Figure 82 illustrates the level of agreement between the tests.

The baseline evaluation results for *quantity* showed an accuracy of 0.9950. The Low EDI score test yielded an accuracy of 0.5025. The High EDI score test demonstrated steady improvement, reaching 95% of baseline accuracy with 20 dimensions, as illustrated in Figure 81. The highest cross-property accuracy was achieved by *negation* and *polarity*, at 0.8525.

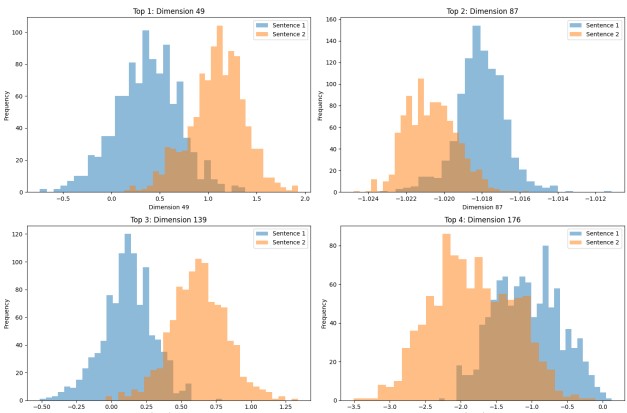

Figure 37: GPT-2 Dimensional Embedding values for the Wilcoxon test results with the most significant p-values for *Factuality*.

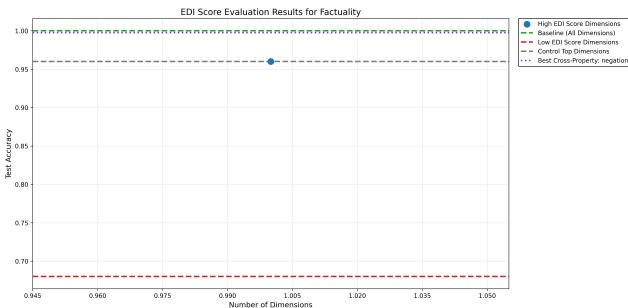

Figure 38: High EDI score evaluation results for GPT-2 Embeddings of *Factuality*.

### F.9 SYNONYM

Figure 83 highlights the difference between the most prominent dimensions encoding this property. Figure 85 illustrates the level of agreement between the tests.

The baseline evaluation results for *synonym* showed an accuracy of $0.6025$. The Low EDI score test yielded an accuracy of $0.4225$. The High EDI score test demonstrated quick improvement, reaching 95% of baseline accuracy with 7 dimensions, as illustrated in Figure 84. The highest cross-property accuracy was achieved by *tense* at $0.5650$.

### F.10 TENSE

Figure 86 highlights the difference between the most prominent dimensions encoding this property. Figure 88 illustrates the level of agreement between the tests.

The baseline evaluation results for *tense* showed an accuracy of $0.9925$. The Low EDI score test yielded an accuracy of $0.5200$. The High EDI score test demonstrated gradual improvement, reaching 95% of baseline accuracy with 17 dimensions, as illustrated in Figure 87. The highest cross-property accuracy was observed with *quantity* at $0.8425$.

### F.11 VOICE

Figure 89 highlights the difference between the most prominent dimensions encoding this property. Figure 91 illustrates the level of agreement between the tests.

The baseline evaluation results for *voice* showed an accuracy of $.9175$. The Low EDI score test yielded an accuracy of $0.3875$. The High EDI score test demonstrated slow improvement, reaching

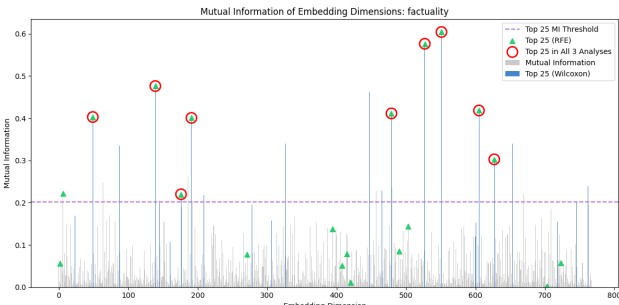

Figure 39: Mutual Information of GPT-2 Embedding Dimensions overlaid with Wilcoxon test and RFE results for *Factuality*.

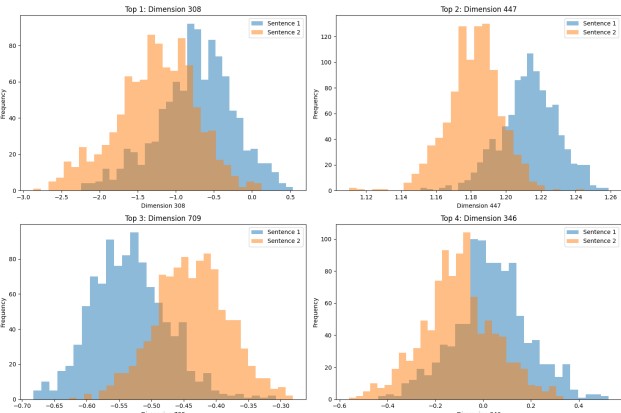

Figure 40: GPT-2 Dimensional Embedding values for the Wilcoxon test results with the most significant p-values for *Intensifier*.

95% of baseline accuracy with 263 dimensions, as illustrated in Figure 90. The highest cross-property accuracy was observed with *definiteness* at 0.6225.

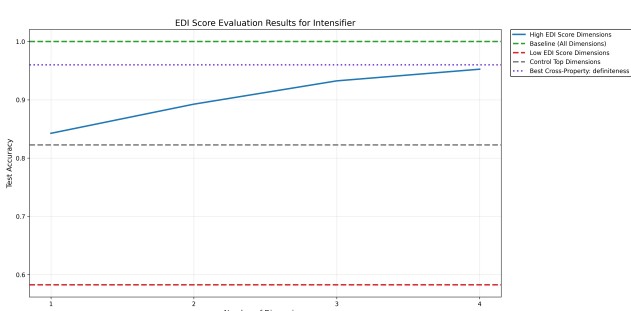

Figure 41: High EDI score evaluation results for GPT-2 Embeddings of *Intensifier*.

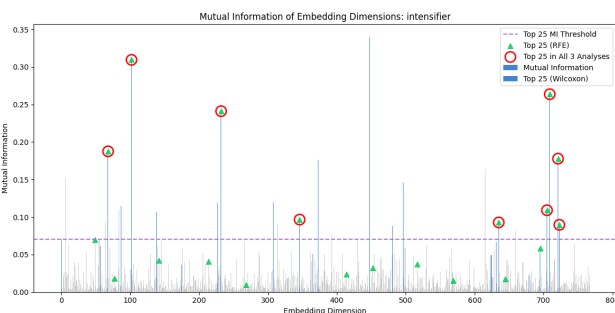

Figure 42: Mutual Information of GPT-2 Embedding Dimensions overlaid with Wilcoxon test and RFE results for *Intensifier*.

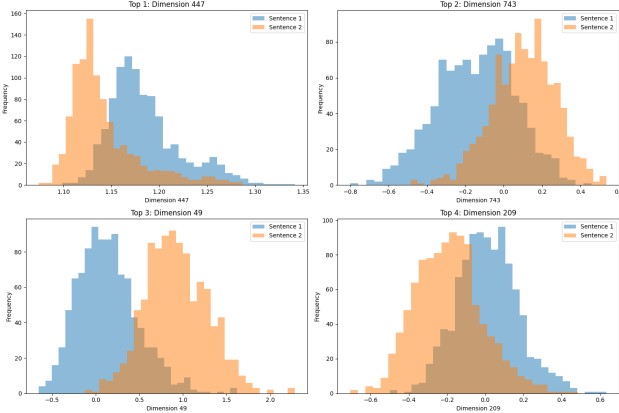

Figure 43: GPT-2 Dimensional Embedding values for the Wilcoxon test results with the most significant p-values for *Negation*.

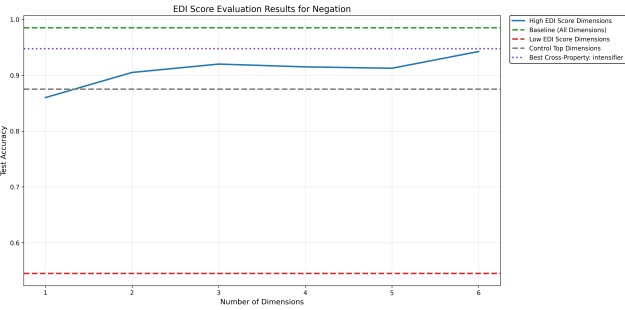

Figure 44: High EDI score evaluation results for GPT-2 Embeddings of *Negation*.

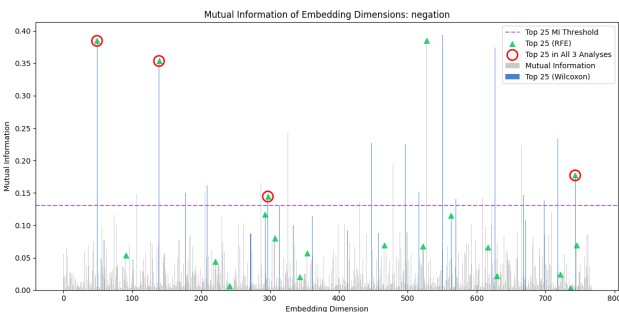

Figure 45: Mutual Information of GPT-2 Embedding Dimensions overlaid with Wilcoxon test and RFE results for *Negation*.

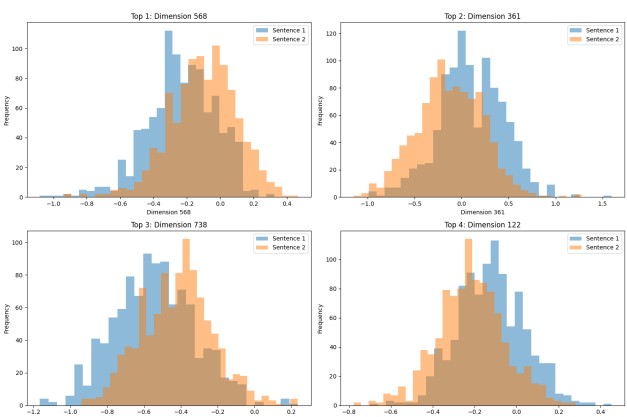

Figure 46: GPT-2 Dimensional Embedding values for the Wilcoxon test results with the most significant p-values for *Polarity*.

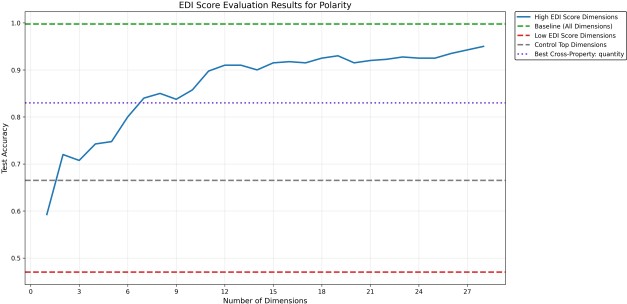

Figure 47: High EDI score evaluation results for GPT-2 Embeddings of *Polarity*.

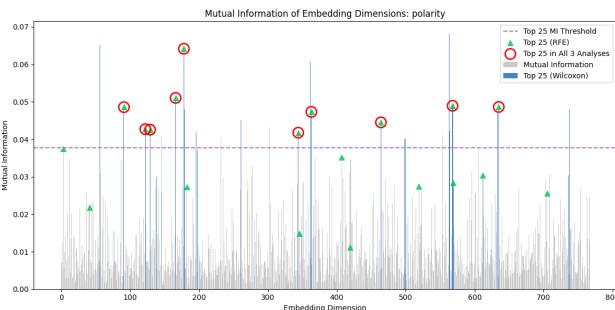

Figure 48: Mutual Information of GPT-2 Embedding Dimensions overlaid with Wilcoxon test and RFE results for *Polarity*.

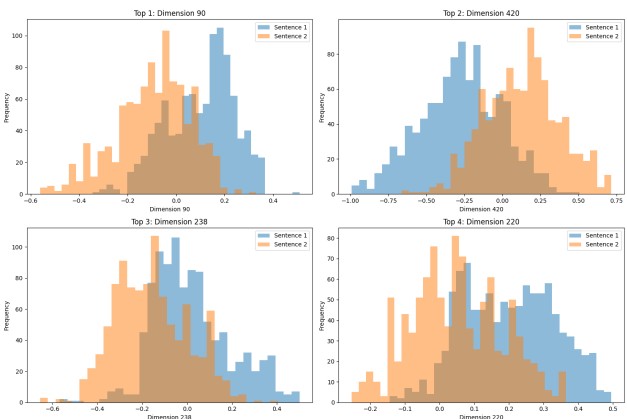

Figure 49: GPT-2 Dimensional Embedding values for the Wilcoxon test results with the most significant p-values for *Quantity*.

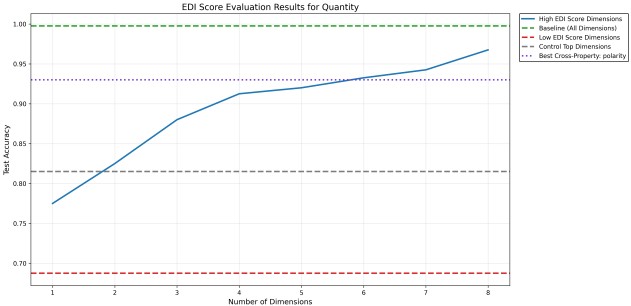

Figure 50: High EDI score evaluation results for GPT-2 Embeddings of *quantity*.

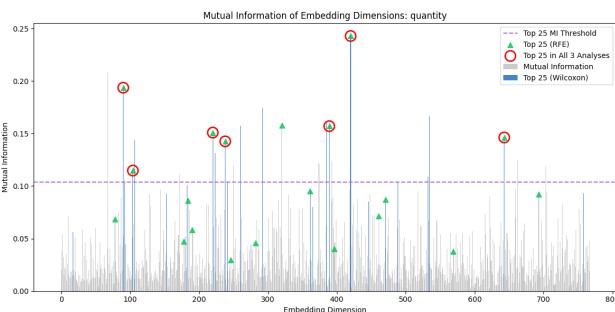

Figure 51: Mutual Information of GPT-2 Embedding Dimensions overlaid with Wilcoxon test and RFE results for *Quantity*

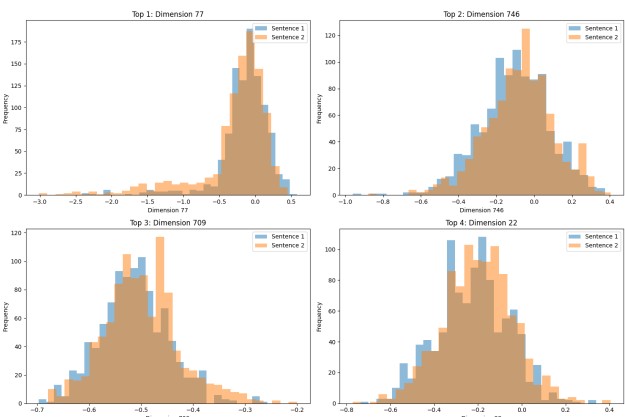

Figure 52: GPT-2 Dimensional Embedding values for the Wilcoxon test results with the most significant p-values for *Synonym*.

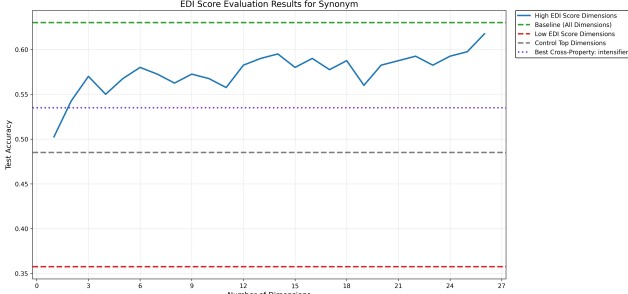

Figure 53: High EDI score evaluation results for GPT-2 Embeddings of *Synonym*.

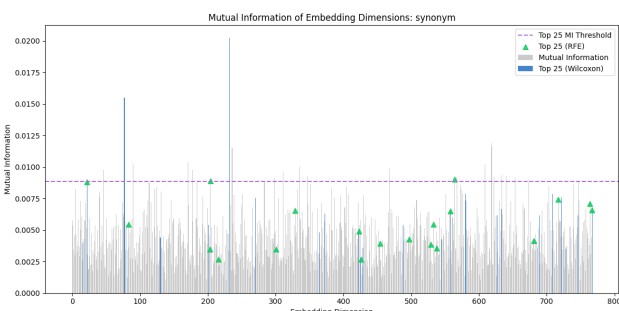

Figure 54: Mutual Information of GPT-2 Embedding Dimensions overlaid with Wilcoxon test and RFE results for *Synonym*.

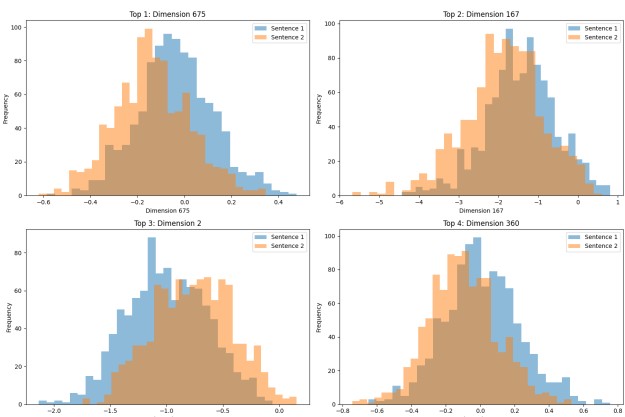

Figure 55: GPT-2 Dimensional Embedding values for the Wilcoxon test results with the most significant p-values for *Tense*.

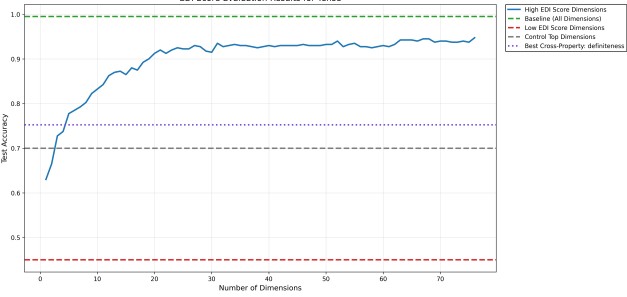

Figure 56: High EDI score evaluation results for GPT-2 Embeddings of *Tense*.

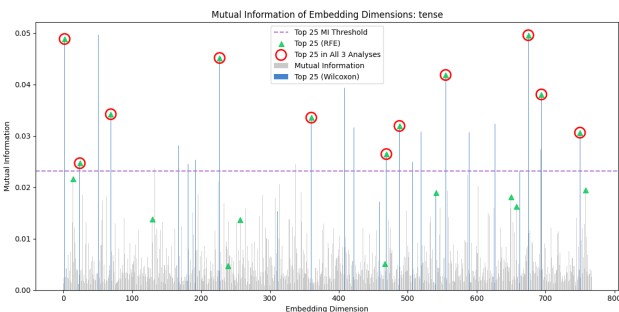

Figure 57: Mutual Information of GPT-2 Embedding Dimensions overlaid with Wilcoxon test and RFE results for *Tense*.

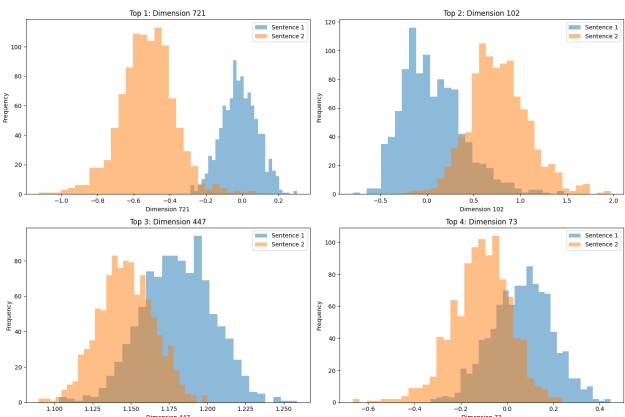

Figure 58: GPT-2 Dimensional Embedding values for the Wilcoxon test results with the most significant p-values for *Voice*.

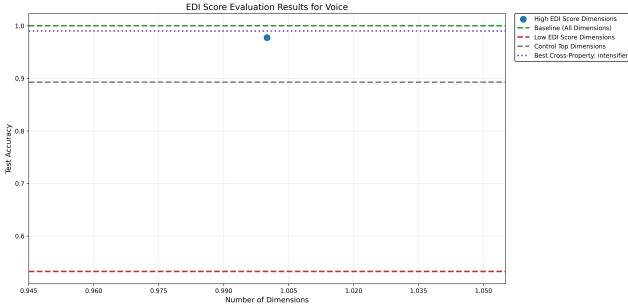

Figure 59: High EDI score evaluation results for GPT-2 Embeddings of *Voice*.

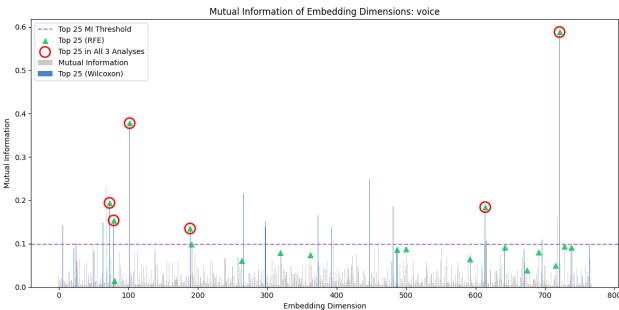

Figure 60: Mutual Information of GPT-2 Embedding Dimensions overlaid with Wilcoxon test and RFE results for *Voice*.

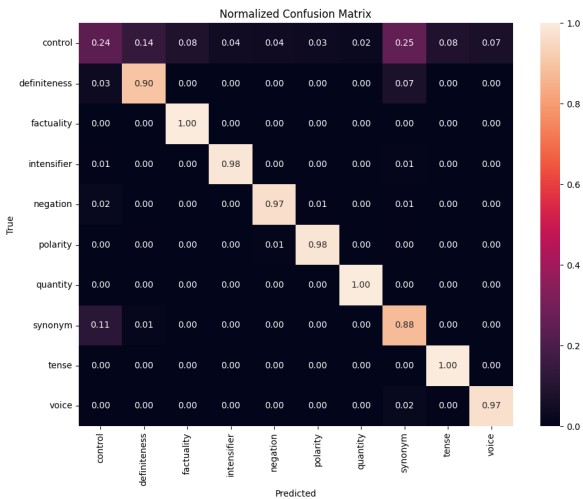

Figure 61: Linguistic Property Classifier results for MPNet.

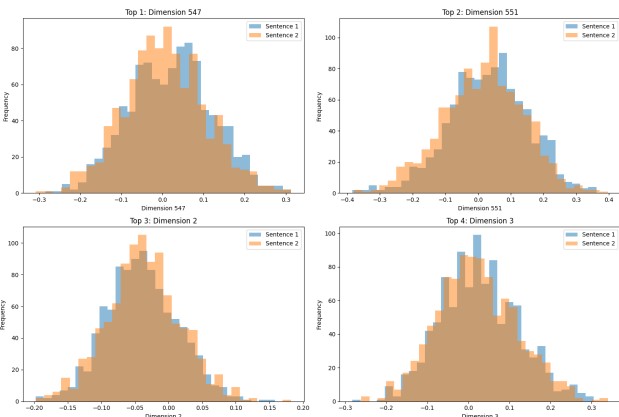

Figure 62: MPNet Dimensional Embedding values for the Wilcoxon test results with the most significant p-values for *Control*.

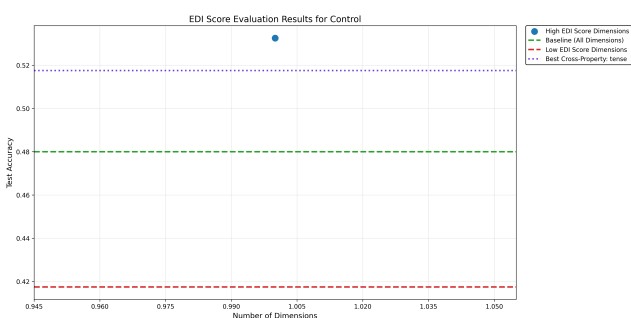

Figure 63: High EDI score evaluation results for MPNet Embeddings of *Control*.

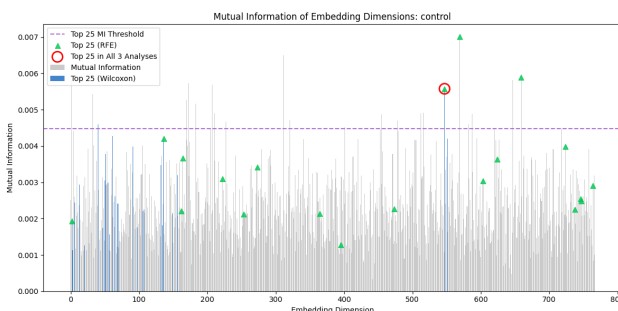

Figure 64: Mutual Information of MPNet Embedding Dimensions overlaid with Wilcoxon test and RFE results for *Control*.

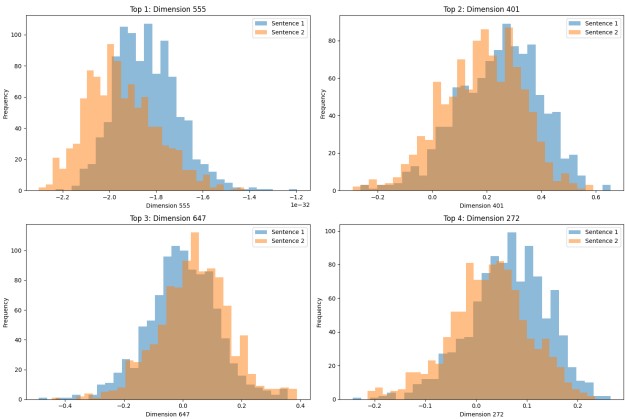

Figure 65: MPNet Dimensional Embedding values for the Wilcoxon test results with the most significant p-values for *Definiteness*.

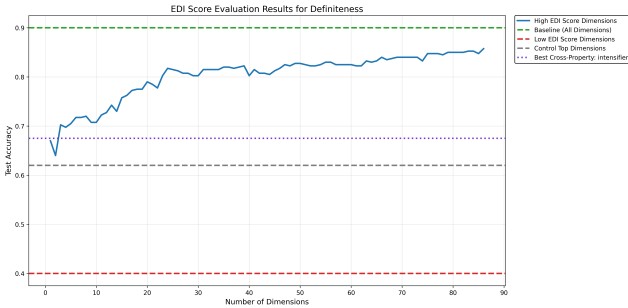

Figure 66: High EDI score evaluation results for MPNet Embeddings of *Definiteness*.

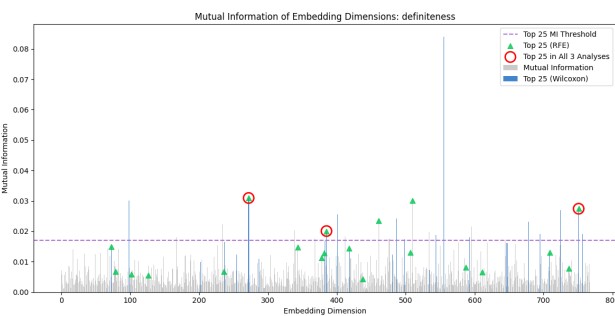

Figure 67: Mutual Information of MPNet Embedding Dimensions overlaid with Wilcoxon test and RFE results for *Definiteness*.

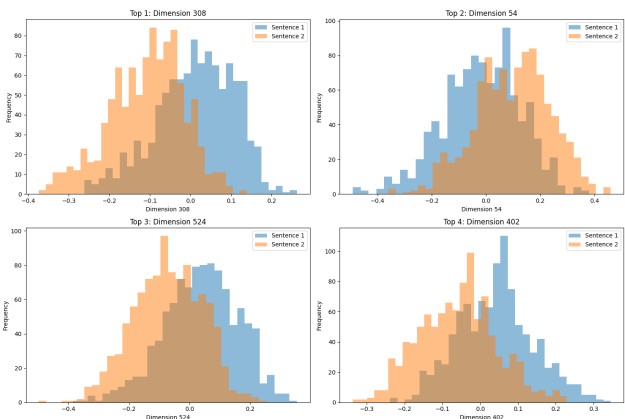

Figure 68: MPNet Dimensional Embedding values for the Wilcoxon test results with the most significant p-values for *Factuality*.

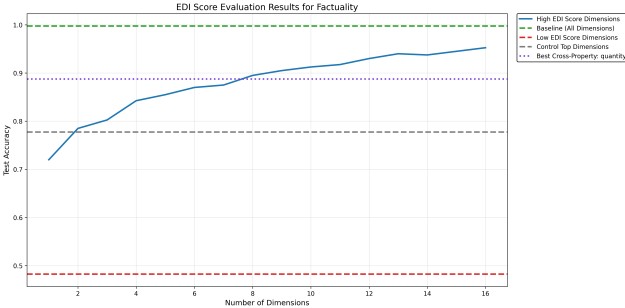

Figure 69: High EDI score evaluation results for MPNet Embeddings of *Factuality*.

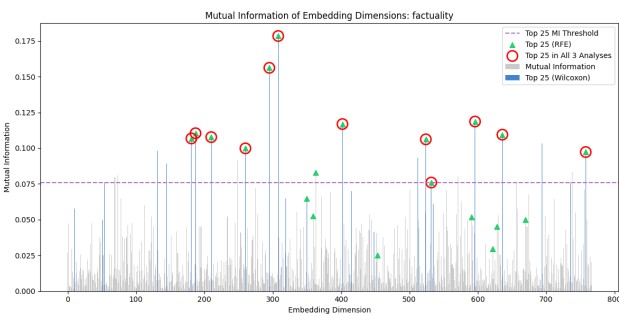

Figure 70: Mutual Information of MPNet Embedding Dimensions overlaid with Wilcoxon test and RFE results for *Factuality*.

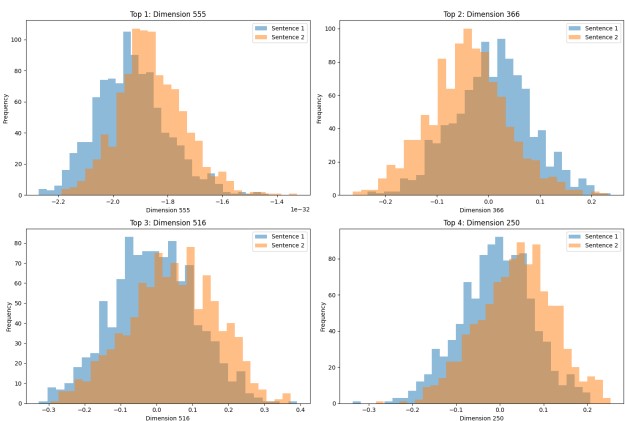

Figure 71: MPNet Dimensional Embedding values for the Wilcoxon test results with the most significant p-values for *Intensifier*.

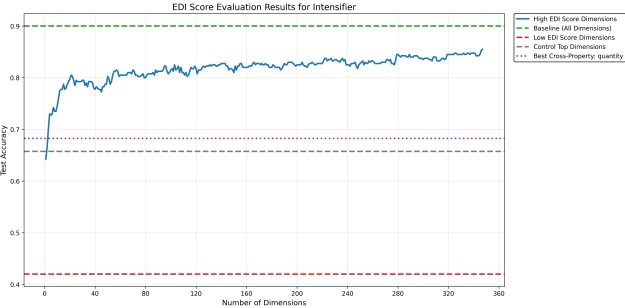

Figure 72: High EDI score evaluation results for MPNet Embeddings of *Intensifier*.

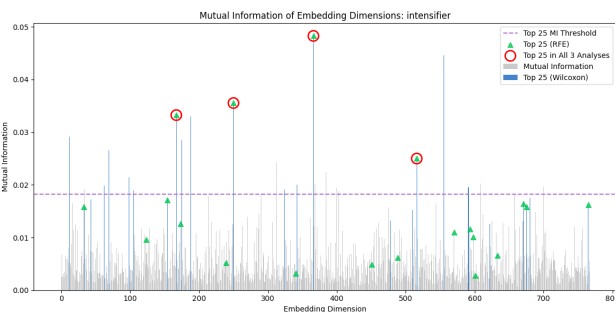

Figure 73: Mutual Information of MPNet Embedding Dimensions overlaid with Wilcoxon test and RFE results for *Intensifier*.

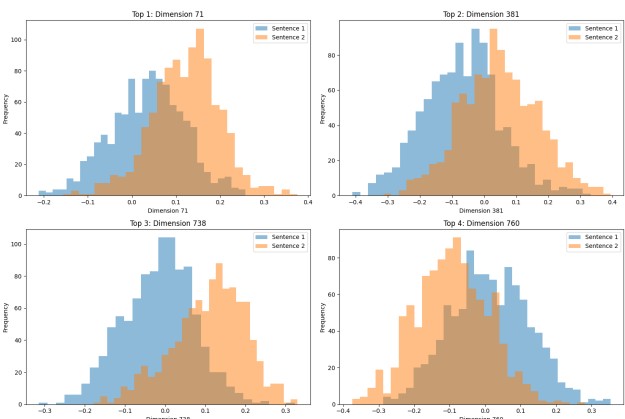

Figure 74: MPNet Dimensional Embedding values for the Wilcoxon test results with the most significant p-values for *Negation*.

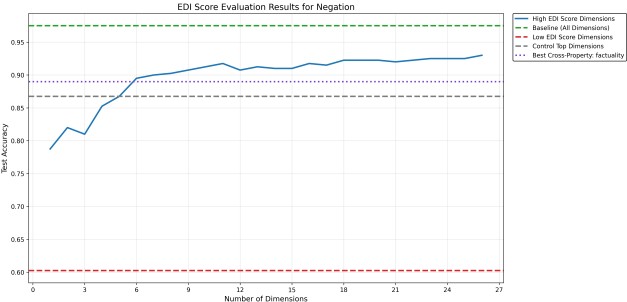

Figure 75: High EDI score evaluation results for MPNet Embeddings of *Negation*.

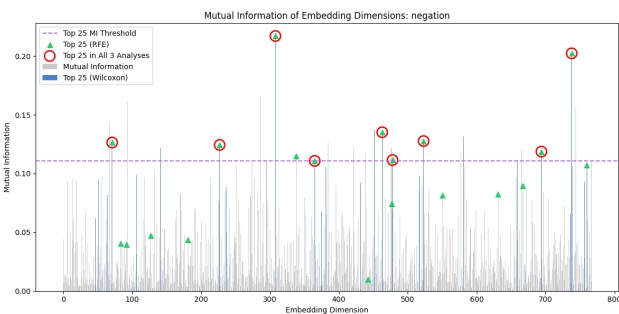

Figure 76: Mutual Information of MPNet Embedding Dimensions overlaid with Wilcoxon test and RFE results for *Negation*.

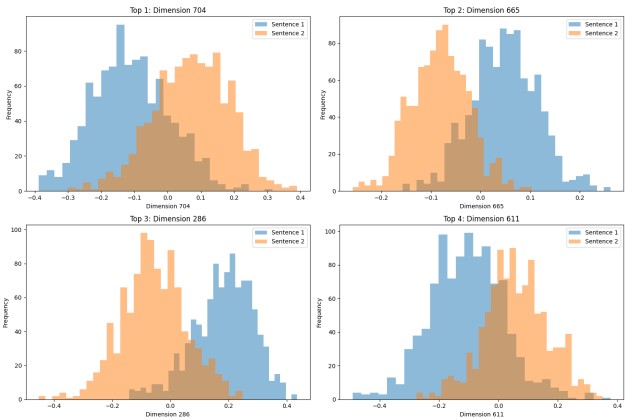

Figure 77: MPNet Dimensional Embedding values for the Wilcoxon test results with the most significant p-values for *Polarity*.

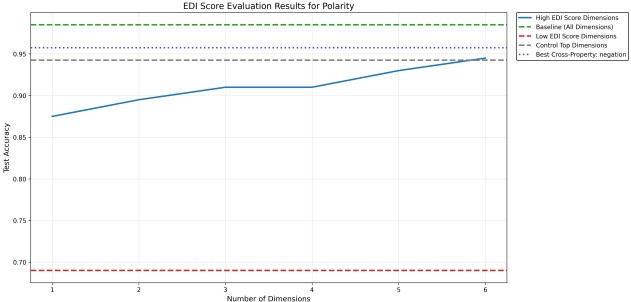

Figure 78: High EDI score evaluation results for MPNet Embeddings of *Polarity*.

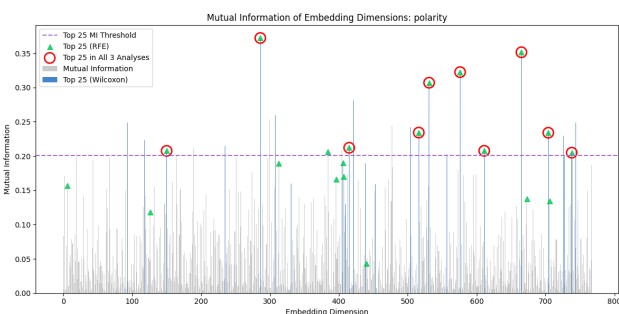

Figure 79: Mutual Information of MPNet Embedding Dimensions overlaid with Wilcoxon test and RFE results for *Polarity*.

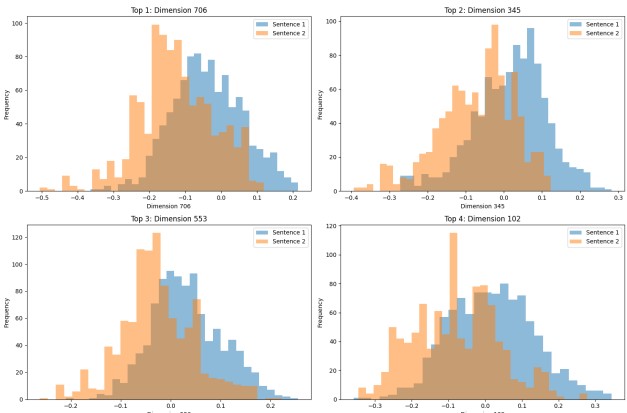

Figure 80: MPNet Dimensional Embedding values for the Wilcoxon test results with the most significant p-values for *Quantity*.

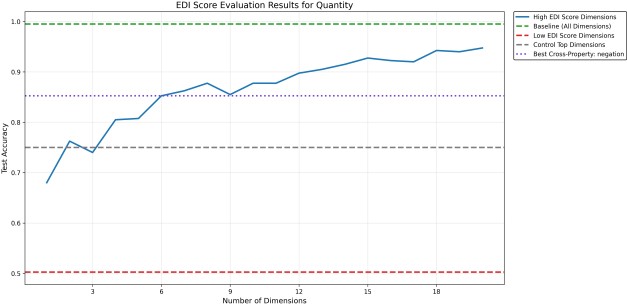

Figure 81: High EDI score evaluation results for MPNet Embeddings of *quantity*.

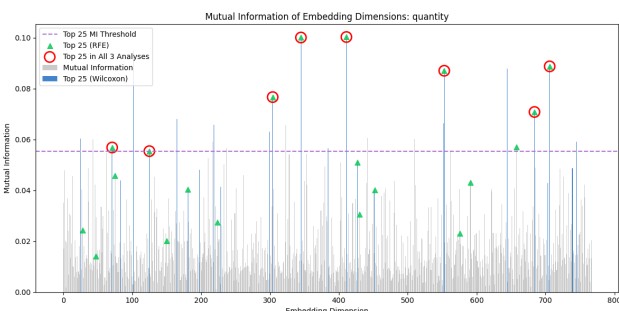

Figure 82: Mutual Information of MPNet Embedding Dimensions overlaid with Wilcoxon test and RFE results for *Quantity*

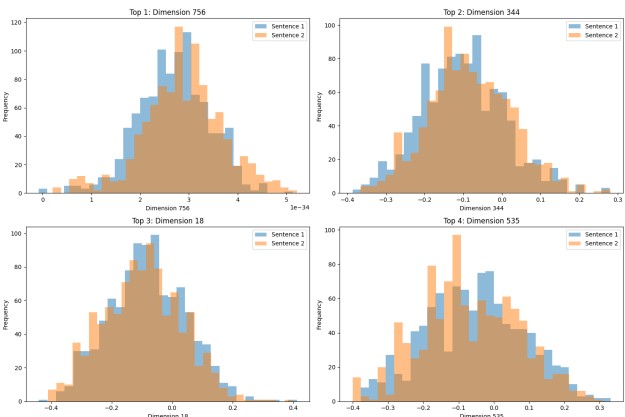

Figure 83: MPNet Dimensional Embedding values for the Wilcoxon test results with the most significant p-values for *Synonym*.

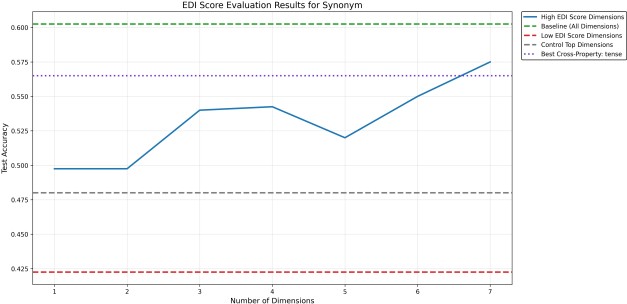

Figure 84: High EDI score evaluation results for MPNet Embeddings of *Synonym*.

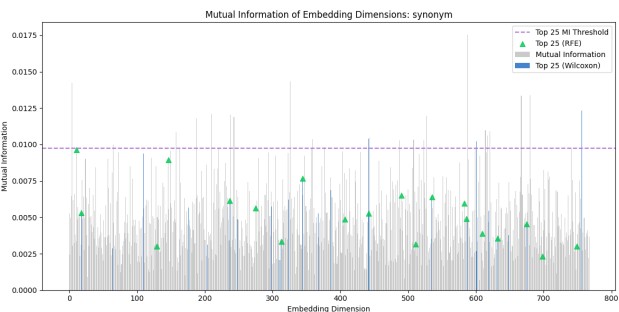

Figure 85: Mutual Information of MPNet Embedding Dimensions overlaid with Wilcoxon test and RFE results for *Synonym*.

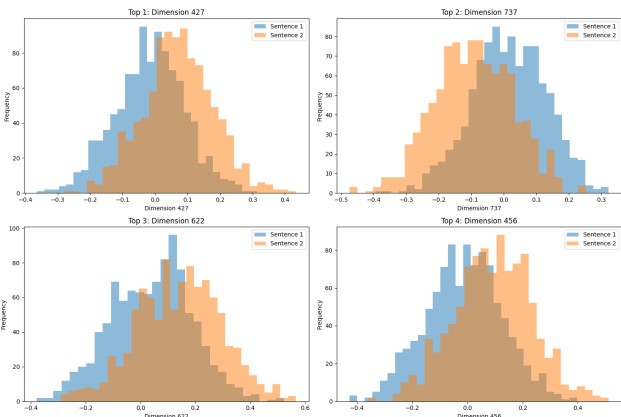

Figure 86: MPNet Dimensional Embedding values for the Wilcoxon test results with the most significant p-values for *Tense*.

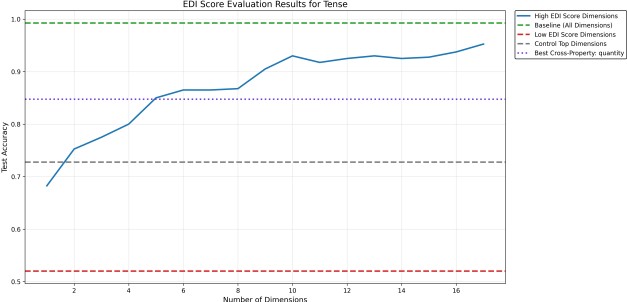

Figure 87: High EDI score evaluation results for MPNet Embeddings of *Tense*.

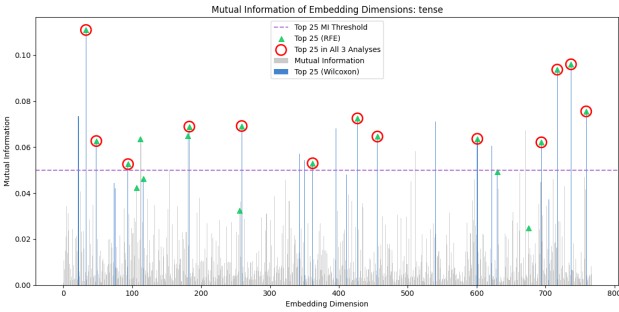

Figure 88: Mutual Information of MPNet Embedding Dimensions overlaid with Wilcoxon test and RFE results for *Tense*.

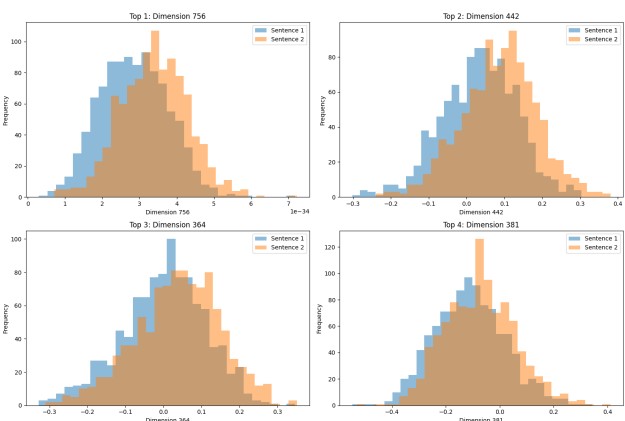

Figure 89: MPNet Dimensional Embedding values for the Wilcoxon test results with the most significant $p$-values for *Voice*.

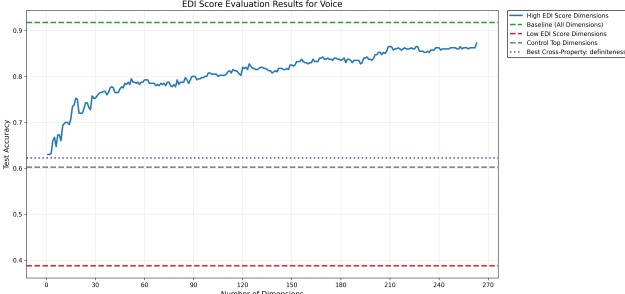

Figure 90: High EDI score evaluation results for MPNet Embeddings of *Voice*.

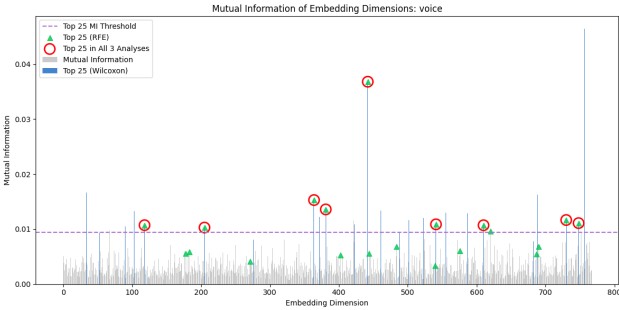

Figure 91: Mutual Information of MPNet Embedding Dimensions overlaid with Wilcoxon test and RFE results for *Voice*.