# OpenReview forum: "Disentangling Linguistic Features with Dimension-Wise Analysis of Vector Embeddings"
_ICLR.cc/2025/Workshop/BuildingTrust — BuildingTrust_

### Official Review · Reviewer_kdQ1 · 2025-02-24

**Rating:** 6
**Confidence:** 4

**Review:**

### Summary
This paper investigates how different linguistic properties (LPs) are encoded in specific dimensions of vector embeddings from models like BERT, GPT-2, and MPNet. The authors introduce the Linguistically Distinct Sentence Pairs (LDSP-10) dataset, designed to isolate ten key linguistic features such as synonymy, negation, tense, and quantity. Using statistical methods, including the Wilcoxon signed-rank test, mutual information, and recursive feature elimination, the paper identifies key embedding dimensions responsible for each LP. It also proposes the Embedding Dimension Importance (EDI) score, a metric for quantifying the relevance of specific dimensions to LPs. The results show that some properties, like negation and polarity, are robustly encoded in distinct dimensions, whereas others, such as synonymy, exhibit more complex patterns. The study contributes to embedding interpretability, bias mitigation, and model optimization.

### Strongness
- Introduces the EDI score to quantify the importance of embedding dimensions for linguistic properties.
- Uses BERT, GPT-2, and MPNet to validate findings, showing consistency across architectures.
- LDSP-10 Dataset – A well-designed dataset that isolates linguistic features, enabling fine-grained analysis of embeddings.

### Weakness
- The connection between Wilcoxon signed-rank test, mutual information, and recursive feature elimination could be explained more clearly, especially how they complement each other in the analysis.
- The study does not include evaluations on LLama or other recent frontier models, so it is unclear how well the findings generalize to the latest architectures.
- The practical applications of the results could be further elaborated, providing more concrete insights into how this analysis can be used in real-world NLP tasks.

---

### Official Review · Reviewer_qciE · 2025-03-02
**The paper effectively explores how specific linguistic features are captured in a subset of embedding dimensions, but it would be even more compelling with token-level analysis and demonstrations of “steering” the model or other theories of impact**

**Rating:** 6
**Confidence:** 4

**Review:**

I appreciate this paper’s attempt to dissect the inner workings of sentence embeddings by zeroing in on specific linguistic properties. The authors clearly put a lot of care into building the LDSP-10 dataset and applying a neat combination of Wilcoxon, mutual information, and recursive feature elimination. It’s interesting to see how a handful of dimensions can capture changes in negation or polarity so strongly. Still, the paper could benefit from token-level analysis to complement the sentence-level approach. That deeper look might reveal local shifts in subwords or single tokens, especially when each linguistic property is introduced.

The writing is generally straightforward, and the approach is explained well enough to follow how LDSP-10 is constructed and how the different statistical methods fit together.

From other interpretability work (e.g., those on superposition), it’s not too surprising that certain properties end up in a low-dimensional subspace. That said, focusing on minimal sentence-pair perturbations is a nice method to highlight which embeddings are doing the “heavy lifting” for each property. The authors do push forward the idea of explicitly measuring how different these dimensions are with multiple tests, which is a solid addition to existing techniques. Overall, the topic is important because knowing where specific properties live in the embedding space can help refine or debug large language models, and perhaps even mitigate biases. It would be even more significant if the authors showed how these identified dimensions might be used in a causal way—actively steering the model’s output by changing embedding values. That would demonstrate a direct application beyond analysis.

Pros and Cons

Pros
There is careful dataset construction in LDSP-10, with minimal changes ensuring a specific linguistic property is isolated. The combination of Wilcoxon, mutual information, and feature elimination is also compelling, especially for confirming certain dimensions matter. The approach highlights how robust negation or polarity signals can be in a relatively small set of dimensions, which is quite informative.

Cons
The paper only provides correlational evidence that a set of dimensions reflect a given property. We don’t see direct intervention experiments, like trying to move the embedding along those dimensions to see if the model changes its behavior accordingly. It also misses the token-level perspective, where local phenomena might be more nuanced. That deeper exploration of direct manipulations or a single-token lens could have made the findings more persuasive.

---

### Official Review · Reviewer_a6aQ · 2025-03-03
**Interpreting Embedding Dimensions**

**Rating:** 7
**Confidence:** 2

**Review:**

The authors curate a dataset and framework for finding features which encode 10 key linguistic properties (LP) including Control, Synonym, Quantity, Tense, Intensifier, Voice, Definiteness, Factuality, Polarity, and Negation. The authors introduce an embedding dimension importance (EDI) metric to measure how entangled an embedding dimension is with an LP. I think the approach is interesting and the conducted experiments verify that these LPs truly exist and are encoded by embedding features. Some suggestions:

1. It's hard to tell which LPs were originally contributed by the authors and which were existent. Could the authors make this clearer early on? Additionally include earlier discussion on how "complete" this list is.

2. It is clear that LPs are very clearly entangled while others have more "complex" relationships. How sure can we be that these 10 LPs are viewed as linearly independent by the model? This is veering closer to mechanistic interpretability/residual streams, but some discussion on this would be interesting.

---

### Decision · Program_Chairs · 2025-03-04

Accept